# High internal phase emulsions gel ink for direct-ink-writing 3D printing of liquid metal

Zewen Lin[1], Xiaowen Qiu[1], Zhouqishuo Cai[1], Jialiang Li[1], Yanan Zhao[1], Xinping Lin[1], Jinmeng Zhang[1], Xiaolan Hu [1] ✉ & Hua Bai [1,2] ✉

3D printing of liquid metal remains a big challenge due to its low viscosity and large surface tension. In this study, we use Carbopol hydrogel and liquid gallium-indium alloy to prepare a liquid metal high internal phase emulsion gel ink, which can be used for direct-ink-writing 3D printing. The high volume fraction (up to 82.5%) of the liquid metal dispersed phase gives the ink excellent elastic properties, while the Carbopol hydrogel, as the continuous phase, provides lubrication for the liquid metal droplets, ensuring smooth flow of the ink during shear extrusion. These enable high-resolution and shape-stable 3D printing of three-dimensional structures. Moreover, the liquid metal droplets exhibit an electrocapillary phenomenon in the Carbopol hydrogel, which allows for demulsification by an electric field and enables electrical connectivity between droplets. We have also achieved the printing of ink on flexible, non-planar structures, and demonstrated the potential for alternating printing with various materials.

Gallium-based liquid metals (LMs), such as gallium-indium and gallium-indium-tin alloys, maintain a liquid state at room temperature and possess high electrical conductivity comparable to conventional metals[1,2]. The combination of liquid state and excellent electrical conductivity make them ideal materials for electrodes in flexible electronic and printed electronic devices[3–6]. Recently, the growing application and advancement of 3D printing technology in flexible electronic devices[7,8] have sparked significant interest in the utilization of LMs for 3D printing[9], particularly the extrusion-based techniques. Extrusion-based 3D printing offers the capability to integrate multiple materials within a single device, enabling the construction of complex three-dimensional circuits[10,11]. This feature is of great appeal for the fabrication of functional devices. However, the low viscosity and high surface tension of LMs pose challenges for their direct application in extrusion-based 3D printing[12,13]. Consequently, controlling the rheological properties of LMs becomes imperative. Currently, the prevailing approach for utilizing LMs in 3D printing involves harnessing the surface oxidation layer that forms spontaneously on these metals when exposed to air. The oxide layer helps to disperse LM droplets within a polymer matrix[14], such as polyvinylalcohol (PVA) solution[15,16], polydimethylsiloxane[17–21], or hydrogel[22], and form composite conductive inks. These composite inks exhibit shear-thinning behavior, facilitating smooth and continuous extrusion, ultimately enabling high-resolution printing.

In fact, these LM-polymer composite inks belong to a dispersed system, and their stability largely relies on the oxide layer formed on the surface of LMs[23]. This oxide layer acts as a barrier that prevents LM droplets from coalescing. However, the mechanical strength of the oxide layer is weak, because it is very thin (0.5–5 nm)[24,25]. During printing, the shear force exerted by the nozzle can potentially cause rupture of the oxide film and subsequent coalescence of LM droplets. This can result in the failure of the printing process and compromise the structural integrity of the printed objects[26]. To avoid the coalescence of embedded LM particles, it is necessary to reduce the volume fractions of LMs to about 50%-75%[16,19,20]. However, the low volume fraction of LM brings other challenges. When the volume fraction of LM is low, the viscosity and modulus of the composite inks are primarily determined by the properties of the polymer matrix. The polymers commonly used for the construction of LM inks, such as uncrosslinked hydrogels or silicone gels[15,16,18–22], generally exhibit low elastic moduli. As a result, while these composite inks may exhibit smooth extrusion through the printing nozzle, the printed lines are

[1]College of Materials, Xiamen University, Xiamen 361005, PR China. [2]Innovation Laboratory for Sciences and Technologies of Energy Materials of Fujian Province (IKKEM), Xiamen, China. ✉e-mail: xlhu@xmu.edu.cn; baihua@xmu.edu.cn

susceptible to creep deformation under the influence of gravity and lack the ability to stack vertically. Consequently, existing LM inks are limited to the printing of two-dimensional (2D) patterns on planar substrates but fail to create stable 3D structures. Thus, there is a contridiction between the printability of the ink and the structural stability of the printed 3D object.

The low content of LM in these composite inks also prevents them from being conductive since the LM droplets are separated by the polymer matrix. To achieve electrical connectivity among embedded LM droplets, external forces or thermal effects are often employed to activate the conductivity of the ink lines. Commonly used methods include pressing[27], stretching[28], laser sintering[29,30], or cryogenic solidification[18,20]. These treatments induce contact between LM droplets, rupture the surface oxide layer, and promote the formation of conductive pathways. However, these approaches are not applicable for most device-level printing. For instance, LM lines embedded within a rigid and opaque substrate cannot be activated by mechanical or laser treatments. Increasing the volume fraction of embedded LM droplets can help to reduce the distance between adjacent droplets, facilitating the rapid percolation of LM throughout the composite inks[31]. However, as mentioned earlier, reducing the distance between adjacent droplets can lead to droplet coalescence during extrusion, resulting in printing failure. Therefore, there is also a contridiction between the printability of LM inks and the activation of conductivity. Further research is needed to explore new methods for achieving high electrical connectivity of LM in composite inks.

In this study, we introduce a Carbopol hydrogel system to prepare LM high internal phase emulsion gel (LM-HIPEG) inks suitable for direct ink writing (DIW). The Carbopol in the hydrogel has a specific interaction with the oxide layer on the LM surface. This interaction enables stable dispersion of high volume fraction (82.5%) LM within the Carbopol hydrogel matrix, resulting in a high internal phase emulsion (HIPE). Based on the structural characteristics of LM-HIPEG, we propose a concept of a lubricant hydrogel layer. We found that the hydrogel layer between LM droplets serves as a lubricant during ink extrusion. This effect reduces the friction force between LM droplets and prevent the oxide layer from rapturing, effectively solving the problem of maintaining the printability while achieving a high volume fraction of LM. This method enables us to successfully print high-resolution, self-supporting 3D LM objects. Furthermore, the polyelectrolyte nature of Carbopol allows for the control of the electric double layer at the Carbopol/LM interface by applying an electric field. This effect, known as electrocapillarity, leads to the development of a method for achieving conductivity activation. By applying low voltage, excellent conductivity can be rapidly achieved in the printed material. Thus, the introduction of Carbopol hydrogel significantly increases the 3D printing performance of LM inks and also provides easy and efficient conductivity activation capability.

## Results

### Preparation and formation mechanism of LM-HIPEG

The present study utilized a Ga-24.5In alloy (in wt.%, referred to as EGaIn) with a low melting point of 16 °C, which remains in a liquid state at room temperature. The preparation process of the high internal phase emulsion gel ink with EGaIn as the dispersed phase and Carbopol hydrogel as the continuous phase is demonstrated in Fig. 1a. Carbopol U20 is a cross-linked copolymer of acrylic acid and $C_{10}$-$C_{30}$ alkyl acrylate, widely used as a rheology-modifying thickener[32,33]. A Carbopol hydrogel can be produced by neutralizing the Carbopol aqueous dispersion with triethanolamine[34-36]. By dispersing EGaIn into Carbopol hydrogel in a volume fraction of 82.5% by simple stirring, a self-supporting ink was obtained, which can maintain stable three-dimensional shapes and does not flow under gravity, as shown in Fig. 1b. In this ink, although the volume fraction of Carbopol hydrogel is as low as 17.5%, it still serves as the continuous phase while EGaIn

becomes the dispersed phase. Since the volume fraction of EGaIn is above the close-packing limit (~74%), the EGaIn droplets come into contact with each other, resulting in mutual squeezing and the formation of polyhedral liquid cells[37]. Eventually, this process leads to the formation of the LM-HIPEG (Fig. 1c). From the transmission microscope image of the ink (Supplementary Fig. 1), it can be observed that the emulsion is of the "liquid metal"-in-water type. The droplets appear non-spherical and display distinct boundaries between them, which are typical features of HIPE. Upon drying, the scanning electron microscopy (SEM) image reveals that the EGaIn droplets form denser polygonal structures (Fig. 1d), and the dried hydrogel film adheres to the surface of the EGaIn droplets (Supplementary Fig. 2, Energy Dispersive Spectrometer images).

We further investigated the mechanism behind the formation of LM-HIPEG ink. Due to the high surface tension of EGaIn, the process of dispersing it into droplets requires the presence of effective surfactants. During dispersion in the matrix, EGaIn comes into contact with air, forming an oxidized surface layer[1,38,39]. As shown in Supplementary Fig. 3, high-resolution transmission electron microscopy (TEM) and energy dispersive spectrometer (EDS) revealed a layer of gallium oxide ~5 nm thick on the surface of EGaIn droplets dispersed in ethanol, which can reduce the surface tension of EGaIn to some extent[40,41], facilitating its dispersion in the matrix. The change in the oxidation state of Ga before and after the formation of LM-HIPEG was analyzed by X-ray photoelectron spectroscopy (XPS)(Supplementary Fig. 4), indicating that a significant amount of oxide layer is generated when EGaIn is dispersed in the Carbopol hydrogel. However, this oxide layer is quite unstable due to its thinness and brittleness. For example, shear dispersion of EGaIn in pure water does indeed result in emulsion formation, but the dispersed EGaIn droplets easily aggregate, leading to demulsification (Supplementary Fig. 5). In the LM-HIPEG system, Carbopol molecules can form strong adsorptive interactions with the oxide layer, significantly improving this instability. During stirring, in addition to oxide layer formation, Ga metal can react with $H_2O$ and oxygen $O_2$ to produce GaOOH, which, upon ionization, releases gallium ions $Ga^{3+}$, $Ga^{3+}$ can coordinate with carboxylate ions, resulting in the adsorption of Carbopol molecules. TEM-EDS line scans (Fig. 1. e-f) show larger amount of C and O elements on the droplet surface compared to those of unmodified LM droplet (Supplementary Fig. 3), confirming the presence of an adsorbed layer of Carbopol molecules. Based on the element distribution, the adsorbed layer is estimated to be about 10 nm thick.

Infrared spectroscopic analysis was employed to further reveal the role of Carbopol hydrogel in the formation of LM-HIPEG (Fig. 1g). For the Carbopol hydrogel, the characteristic bands in the range of 1800 - 1650 $cm^{-1}$ are ascribed to the stretching vibration of carboxyl group ($\upsilon C = O$). The asymmetric and symmetric stretching vibrations of $COO^-$ appear at 1650 - 1500 $cm^{-1}$ and 1430 - 1370 $cm^{-1}$, respectively. EGaIn is dispersed into Carbopol hydrogel through stirring, leading to coordination between carboxylate ions and $Ga^{3+}$ released from GaOOH groups on the surface. Spectroscopic analysis reveals the complete absence of the carboxyl band ($\upsilon C = O$), alongside a blue-shift in the asymmetric stretching vibration band of carboxylate. These changes indicate that all the carboxyl groups of Carbopol chains are coordinated with $Ga^{3+}$ ions. This phenomenon is consistent with existing research on $Ga^{3+}$ interactions with carboxylate complexes, suggesting the formation of inner-sphere coordination between $Ga^{3+}$ and the carboxyl groups in Carbopol[42-44]. The interaction leads to strong adsorption of the hydrogel layer on the surface of EGaIn droplets, further reducing the surface energy and enabling the dispersion of EGaIn droplets in the Carbopol hydrogel. Additionally, the electrostatic repulsion between Carbopol molecular chains increases the solution's viscosity, forming a gel layer. This hydrogel layer acts as a viscoelastic protective layer for EGaIn, providing spatial hindrance and lubrication, making the dispersion more stable. Additionally, the

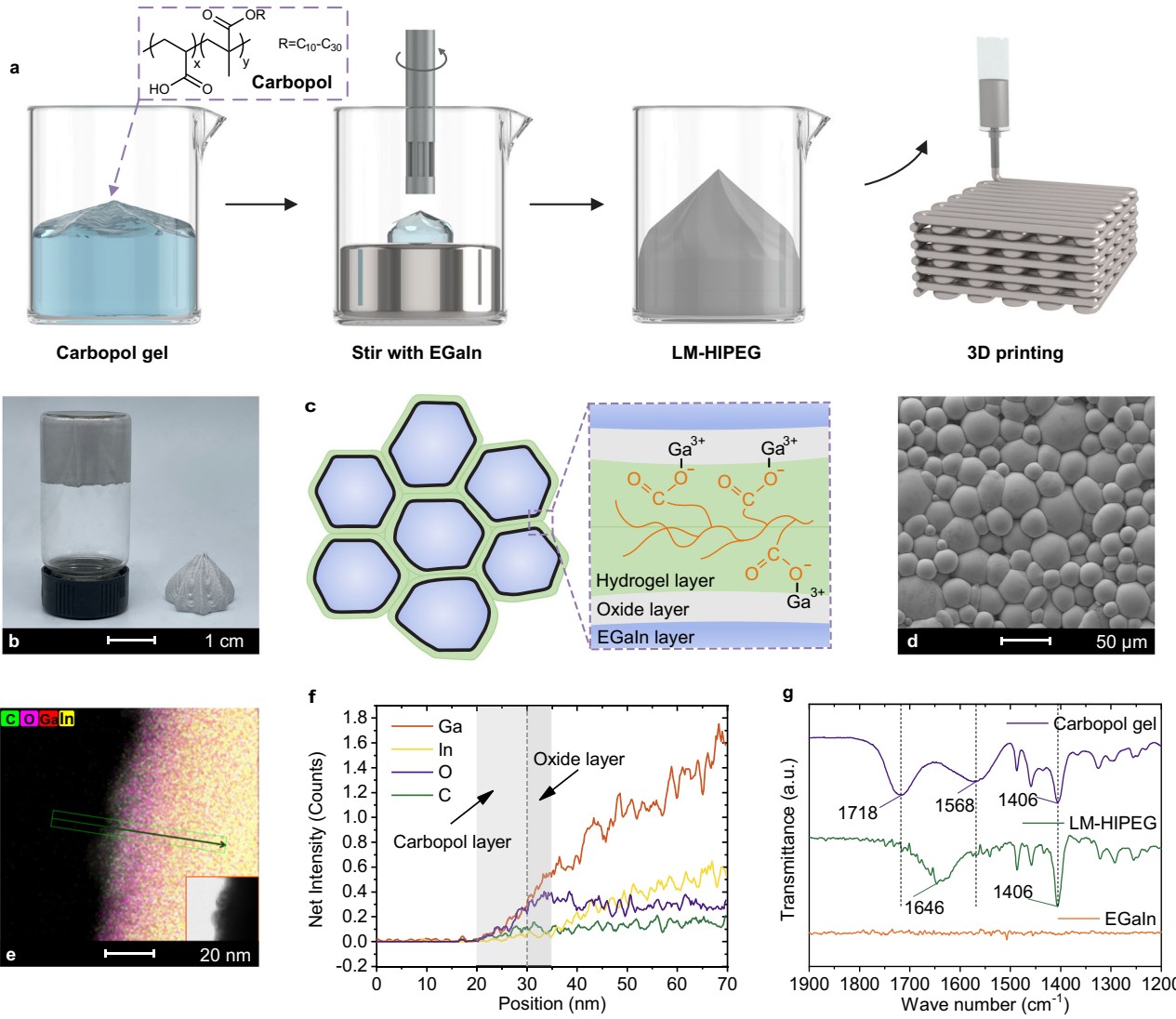

**Fig. 1 | Preparation process and formation principle of LM-HIPEG. a** Schematic diagram of the preparation of LM-HIPEG. **b** Photos of LM-HIPEG. **c** Conceptual illustration of the emulsion structure of LM-HIPEG. **d** SEM morphology of LM-HIPEG. **e, f** TEM/EDS of LM-HIPEG. **g** Infrared spectra of Carbopol hydrogel, LM-HIPEG, and EGaIn.

hydrogel layer also serves as a viscoelastic protective layer. Atomic force microscopy (AFM) force curves reveal that EGaIn droplets with the Carbopol hydrogel layer exhibited higher puncture resistance compared to droplets with only an oxide layer (Supplementary Fig. 6). This means that the LM-HIPEG with Carbopol hydrogel can withstand larger external force without demulsification. Therefore, the gallium oxide layer formed on the surface of EGaIn droplets, along with the adsorbed Carbopol hydrogel layer, serves as the surfactants in LM-HIPEG system, ensuring its dispersion stability.

## Rheological properties of LM-HIPEG

Rheological properties are commonly used to evaluate the suitability of inks for DIW printing, because the moduli of inks play crucial roles in maintaining the desired shape of the printed object. Rheological tests show that the storage modulus (G′) of LM-HIPEG exceeds $10^4$ Pa, which is two orders of magnitude higher as compared to pure Carbopol hydrogel (Supplementary Fig. 7 and Fig. 2a). Meanwhile, G′ surpasses the corresponding loss modulus (G″) by one order of magnitude, suggesting that the LM-HIPEG ink primarily exhibits elastic properties, enabling it to maintain its shape. The elasticity of LM-HIPEG arises from the fluid nature of the dispersed EGaIn droplets, which can store energy by the deformation under applied stress, and it does not have a

direct correlation with the modulus of the continuous phase (Supplementary Fig. 8). The elastic modulus of a HIPE is determined by the effective volume fraction of the dispersed phase ($\varphi_{eff}$), interfacial tension (σ), and the radius of the droplets (r). The size distribution of EGaIn droplets tends to remain consistent across different volume fractions of LM-HIPEG after prolonged shear dispersion (Fig. 2b and Supplementary Fig. 9). As a result, higher volume fractions of EGaIn in the ink result in a greater elastic modulus, as shown in Fig. 2a. For example, increasing the volume fraction of EGaIn from 77.5% to 85% results in a rise in the initial storage modulus plateau from 11,500 Pa to 41,000 Pa. Based on the test results from Fig. 2a, the relationships between the EGaIn volume fraction and both the energy storage modulus and the yield stress can be fitted respectively (Supplementary Fig. 10) as: $G′ \sim \varphi_{eff} (\varphi_{eff} - 1.51) \Delta P$, $\tau_y \sim \varphi_{eff} (\varphi_{eff} - 1.43) \Delta P$, where $\Delta P$ represents the Laplace stress ($\Delta P = 2\gamma/r$). The fitting results are both close to the conclusions of previous literature[45,46]. The results of the Three Interval Thixotropy Test (3ITT) test indicate that the ink's modulus is capable of rapidly responding and recovering as it transitions from a high shear to a low shear stage (Supplementary Fig. 11). Comparing the ink modulus and volume fraction of LM with literature data[15,16,18–20,47] (Fig. 2c), the data point in this study falls in the upper right corner of the comparison chart, indicating that the ink has high

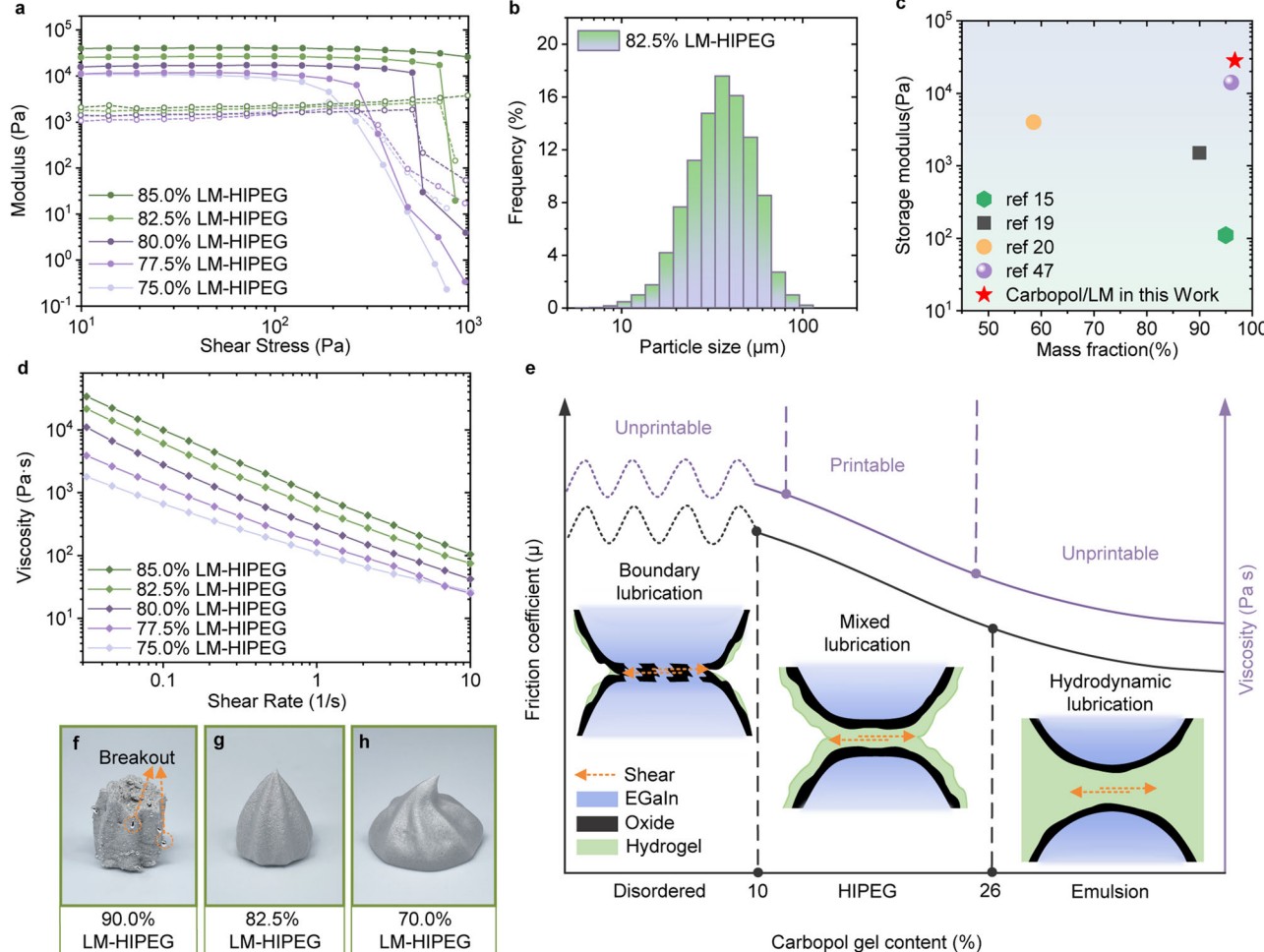

**Fig. 2 | Rheological properties of LM-HIPEG. a** Modulus as a function of shear stress for LM-HIPEG with different volume fractions of EGaIn (75% ~ 85%). The solid line represents the stored energy modulus, and the dashed line represents the loss modulus. **b** Particle size distribution of LM-HIPEG. **c** Comparison of storage modulus and volume fraction of LM-HIPEG with literature data[15,19,20,47]. **d** Viscosity as a function of shear rate for LM-HIPEG with different volume fractions of EGaIn (75% ~ 85%). **e** Three different lubrication effects of LM-HIPEG system with different volume fractions of Carbopol hydrogel. **f-g** Morphology of LM-HIPEG with an EGaIn volume fraction of 90% (**f**), 82.5% (**h**), and 70% (**g**).

storage modulus, which is beneficial for high resolution, self-supportability, and shape stability during printing.

Shear-thinning behavior is crucial for ink extrusion printing technology. Figure 2d demonstrates that the viscosity of ink, with a volume fraction of LM ranging from 77.5% to 85%, exhibits typical shear-thinning behavior. During the flow of LM-HIPEG, the relative motion of EGaIn droplets against each other leads to squeezing and friction. As a result, the viscosity of LM-HIPEG is higher than that of pure Carbopol hydrogel, as shown in Supplementary Fig. 12. As the shear force increases, the hydrogel network of the continuous phase is disrupted, leading to a decrease in viscosity. At the same time, the droplets are stretched by the shear force and align with the direction of the shear force, reducing the squeezing between particles. These two factors combined contribute to the shear-thinning rheological characteristics of the ink.

It is important to emphasize that Carbopol hydrogel plays a vital role in enabling the printability of the LM-HIPEG ink. As mentioned above, a high volume fraction of LM is required to maintain the shape stability of the ink under gravity. However, due to the small thickness of the oxide layer on the surface of EGaIn droplets, the oxide layer is prone to rupture from the friction and squeezing between droplets during the extrusion process. Once the oxide layer is breached, EGaIn will escape and coalesce, resulting in demulsification. Hence, there is a

trade-off between shape stability and extrusion stability. Nevertheless, Carbopol hydrogel acts as a lubricant and protector for EGaIn droplets, ensuring the stability of the emulsion during ink flow. The lubricating effect of Carbopol hydrogel in LM-HIPEG can be categorized into three types based on the Stribeck curve[48,49], depending on the thickness of the hydrogel between LM droplets (Fig. 2e): (1) boundary lubrication, (2) mixed lubrication, and (3) fluid lubrication. Boundary lubrication occurs when the quantity of Carbopol gel is insufficient to completely encapsulate the droplets, resulting in sufficient contact between the droplets, and consequently higher friction coefficients. Under shear forces, the oxide layers experience frictional rupture, causing EGaIn to coalesce. As shown in Fig. 2f, when the volume fraction of EGaIn reaches 90%, the ink enters the boundary lubrication region. During the extrusion of the ink from a 7 mm diameter tube, demulsification occurs, resulting in the formation of EGaIn droplets. When the volume fraction of EGaIn is between 74% and 90%, the ink exhibits mixed lubrication. This means that the adjacent LM droplets are separated by a thin layer of adsorbed hydrogel, with the calculated thickness of this gel layer ranging from 2.38 to 1.33 μm(Supplementary Fig. 13). Although the droplets still make surface contact with each other, the primary friction occurs between the absorbed gel layers, resulting in moderate frictional forces. This allows the oxide layer on the surface of LM droplets to remain intact during printing (Fig. 2g). The rheological

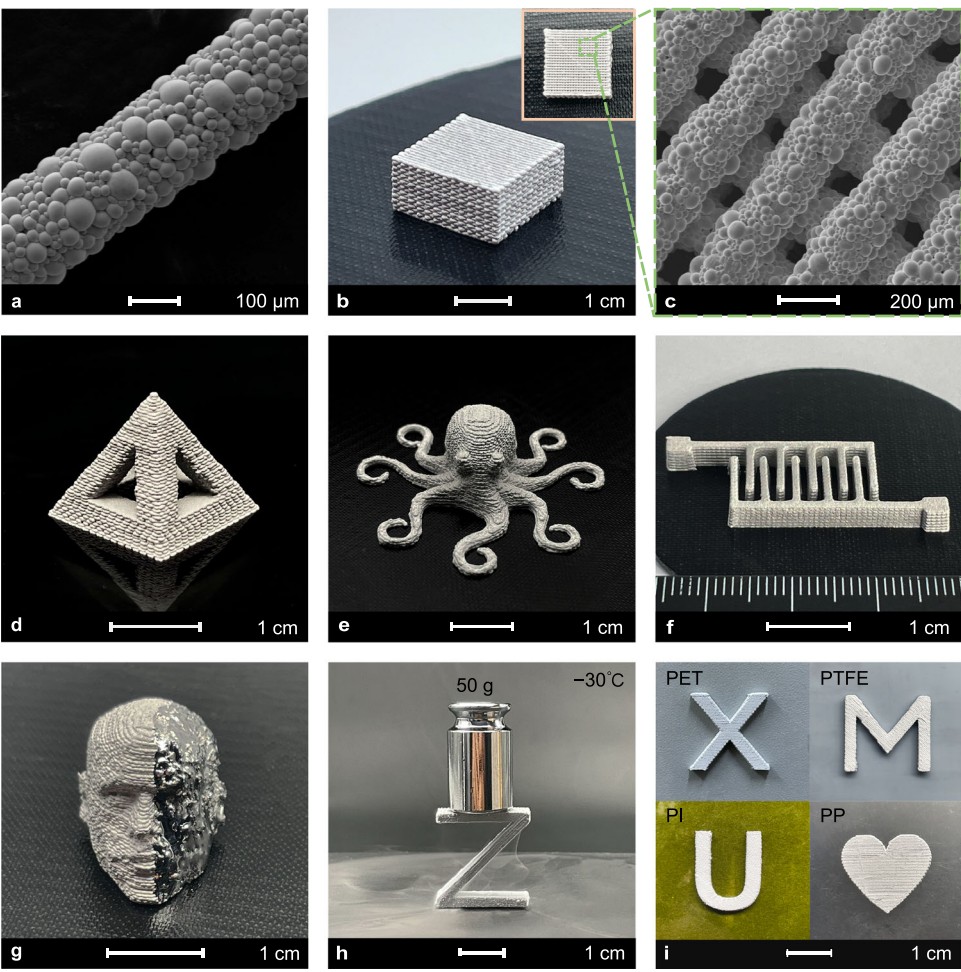

**Fig. 3 | 3D Printing of LM-HIPEG. a** SEM image of a single printed line. **b** Photo of a printed non-densely filled cubic (**c**) SEM image of the network of printed filaments in (**b**). **d** Photo of a printed tetrahedron with a hollow structure. **e** Photo of a printed Octopus object. **f** Photo of a pair of printed interdigital electrodes. **g** Photo of a printed human head object, the left part of which was scraped to release LM. **h** Photo of a zigzag structure solidified at −30 °C supporting a weight. **i** Photos of X, M, U, and heart-shaped patterns printed on PET, PTFE, PI, and PP substrates.

properties and particle size distribution of the 82% LM-HIPEG ink after extrusion remained virtually unchanged, and there was no residue found in the nozzle after extrusion (Supplementary Fig. 14). These inks have both good shape stability and extrusion performance, thereby suitable for printing. When the EGaIn volume fraction is <74%, the ink transitions from a high internal phase emulsion to a conventional emulsion. At this point, the lubricating effect of Carbopol hydrogel changes to fluid lubrication type. In this state, the droplets no longer make contact with each other, and Carbopol hydrogel fills the interparticle space. The friction coefficient is very low and primarily depends on the viscosity of Carbopol hydrogel. Although the ink still shows shear-thinning behavior and is easy to extrude, the lower EGaIn content leads to decreased energy storage modulus and reduced self-supporting ability (Fig. 2h, See Supplementary Fig. 15 for the state of other inks), which is unfavorable for 3D printing. At the same volume fraction of EGaIn, we compared composite inks made from liquid metal and several other polymeric matrix materials, such as poly-dimethylsiloxane (PDMS), polyethylene oxide (PEO) hydrogel, and polyvinyl alcohol (PVA) solution. As illustrated in Supplementary Fig. 16, these materials either failed to form stable emulsions or experienced severe emulsion breaking during stirring/extrusion, due to the lack of functional groups that interact with the oxide layer of the liquid metal droplets or the absence of a gel layer that acts as a lubricant during shearing. In addition, the Carbopol gel layer will serve as

the apparent slip layer of the ink, directly interfacing with the plastic syringe and the stainless steel printing needle. A thicker Carbopol gel layer can create bands that do not transfer shear stress effectively, resulting in beneficial wall slip-type shear bands during the printing process, which avoid damage to the oxide layer. Therefore, Carbopol hydrogel plays an important role in ensuring excellent shear-thinning performance of the ink at high LM contents. In our subsequent studies, we used ink with a volume fraction of 82.5% EGaIn.

## 3D Printing of LM-HIPEG

The ink's high viscoelasticity and shear-thinning rheological properties make LM-HIPEG highly suitable for being printed by DIW 3D printing into high-resolution, high-aspect-ratio objects. The shear-thinning nature of the inks enables it to be printed with a 200 μm diameter nozzle. The SEM image in Fig. 3a shows that the printed lines have a diameter of 210 μm and exhibit minimal extrusion swelling, which enhances the printing accuracy. The lines are formed by closely packed EGaIn droplets. Since water is the only volatile solvent in LM-HIPEG, the surface of the printed lines and structures can dry in ambient conditions quickly. However, because EGaIn is not permeable to water(-Supplementary Fig. 17), the evaporation process of water within the ink is significantly slowed down, which means that after printing, the printed lines undergo minimal shrinkage and maintain their cylindrical shape. The printed 3D objects can remain stable for up to 3 h in

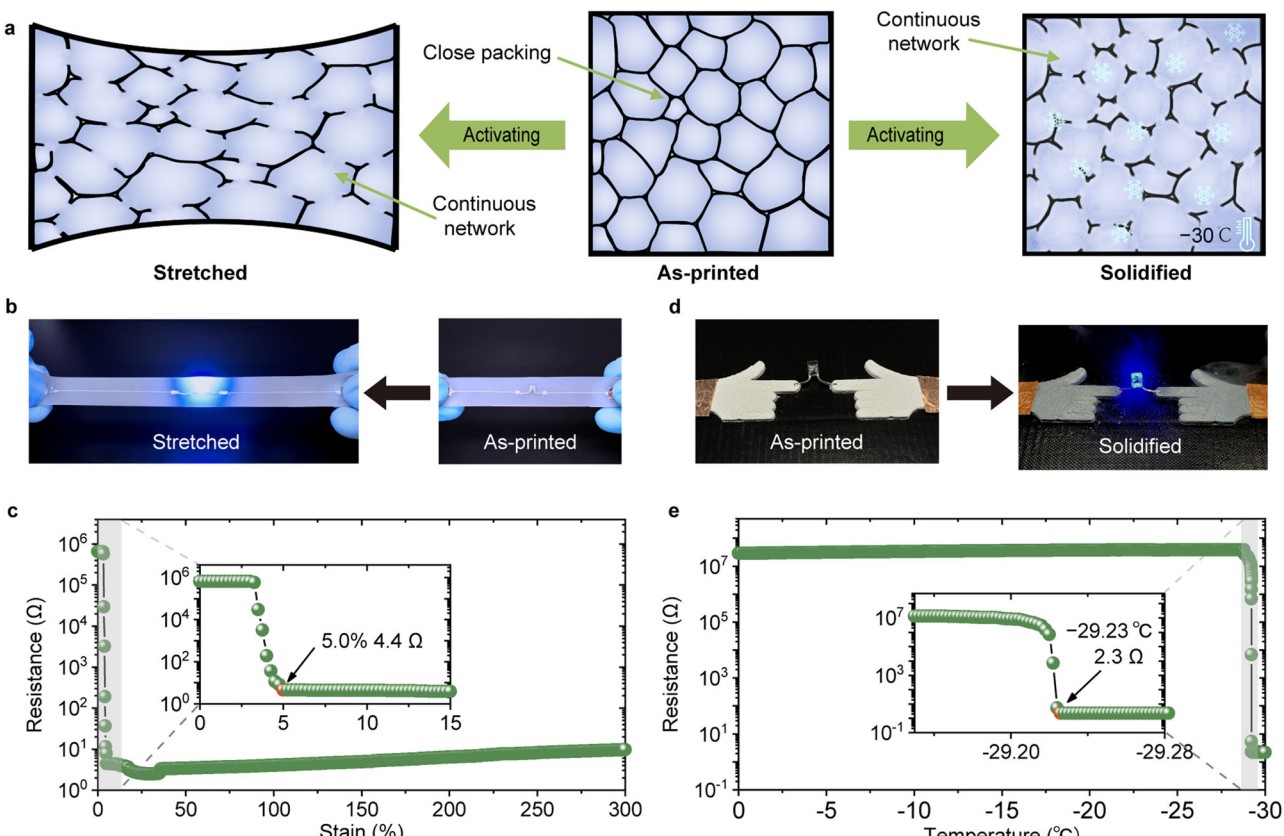

**Fig. 4 | Strain- and solidification-induced conductivity activation of LM-HIPEG.** **a** Schematic illustration of strain-activation (left part) and solidification-activation (right part) process of LM-HIPEG. **b** Photos of a printed LM-HIPEG line in an LED circuit before and after strain-induced conductivity activation. **c** Resistance variation of printed line under 300% strain. **d** Photos of a printed LM-HIPEG line in an LED circuit before and after solidification-induced conductivity activation. **e** Resistance variation of a printed line as the temperature decreases from 0 – −30 °C.

ambient conditions, while 2D patterns can be preserved for over 24 h at 100 °C (Supplementary Fig. 18). The easy merging of liquid metal droplets facilitates subsequent activation of the ink's conductivity while also leading to shape changes. Figure 3b, c shows a 2 cm × 2 cm × 1 cm cubic structure with interlayer staggered, non-dense filling, demonstrating high resolution and accuracy of the printing. LM-HIPEG exhibits excellent designability and printability, making it suitable for functional device fabrication and artistic creations. Examples of printed objects include a hollow tetrahedron, an octopus model, and a pair of interdigital electrodes (Fig. 3d–f), highlighting the precise shape and high resolution of the printing. Supplementary Movie 1 provides a detailed view of the printing process and fine details. It is worth noting that the hollow tetrahedron features a large-span suspended structure, thus showcasing the high shape stability of the ink.

The EGaIn can easily escape from the oxide layer under the action of external forces, transforming the printed object into a liquid state. Figure 3g demonstrates a human head model where the left part surface was turned into liquid using a scraper. On the other hand, 3D printing with LM-HIPEG can meet the structural design requirements for loaded components under low-temperature conditions. At −30 °C, the LM-HIPEG 3D printed object solidifies. The Z-shaped solidified 3D printed object, with a cross-sectional area of 12 mm², can support a 50 g weight, showing a certain load-bearing capacity (Fig. 3h). Furthermore, LM-HIPEG ink exhibits good interfacial compatibility with common polymer materials, allowing for the direct construction of complex 3D LM patterns on these flexible substrates, such as polyethylene terephthalate (PET), polytetrafluoroethylene (PTFE), polyimide (PI), and polypropylene (PP) (Fig. 3i). This good compatibility can be attributed to the present of

Carbopol hydrogel on the surface of EGaIn droplets, which contains both hydrophilic (−COOH) and hydrophobic (−C = O) functional groups. In contrast, pure EGaIn does not wet these materials well (Supplementary Fig. 19). This feature facilitates the synergistic printing of LM ink with other polymer inks, enabling the construction of complex devices. These findings collectively demonstrate the excellent 3D printability of LM-HIPEG ink, which will propel the advancement of LM-based printing and flexible device design toward high-resolution three-dimensional structures.

## Conductivity activation of LM-HIPEG

From the SEM image (Fig. 3a), it can be observed that LM droplets in LM-HIPEG are densely packed but separated by oxide and hydrogel layers. These layers are non-conductive, so the ink is not electrically conductive. However, when subjected to strain, the oxide shell and hydrogel layer can rupture, causing the partial coalescence of the EGaIn droplets and forming a conductive pathway[15] (Fig. 4a, left part, and Fig. 4b). Here, a line with a diameter of 210 μm was printed on a PDMS substrate, and during the stretching process, significant variations in the relationship between PDMS strain and conductivity of the line were observed (Fig. 4c). Notably, the resistance of the line decreases from 657.6 kΩ to 4.4 Ω within a 5% strain, (Supplementary Movie 2), resulting in an impressive conductivity of $2.6 × 10^5$ S/m. As the line is further stretched to a strain of 300%, the resistance remains within the range of 2.5-9.7 Ω. Additionally, the resistance of the line exhibits a consistent and proportional change with strain, demonstrating excellent cyclic stability (Supplementary Fig. 20). This activation method is suitable for the LM-HIPEG patterns on flexible substrates.

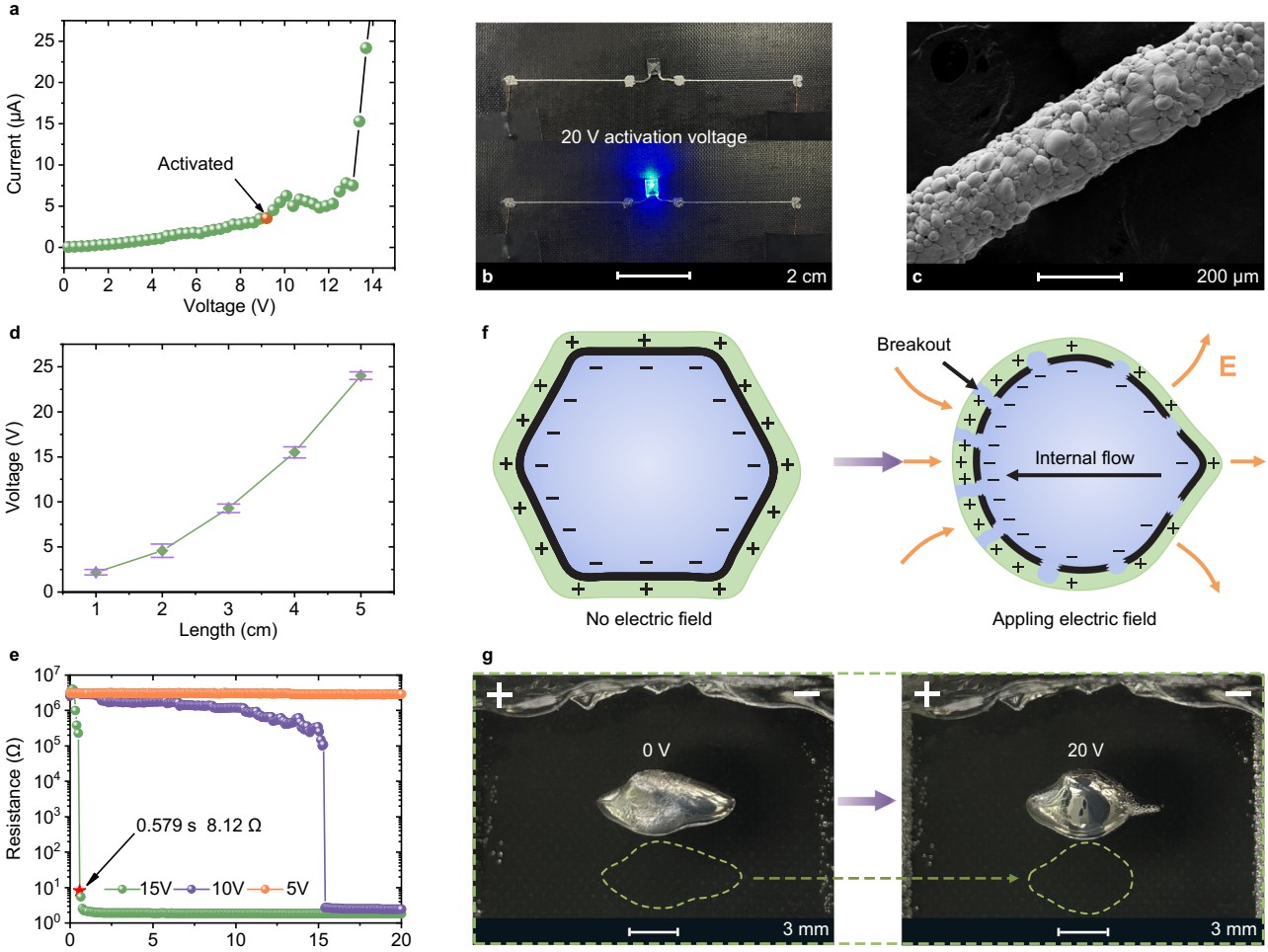

**Fig. 5 | Voltage-induced conductivity activation of LM-HIPEG. a** Current-voltage curve of an as-printed line. **b** Photos of a printed LM-HIPEG line in an LED circuit before and after voltage-induced conductivity activation. **c** SEM image of printed lines after voltage-induced activation. **d** Comparison of activation voltages for printed lines with different lengths (Error bars represent standard deviation, $n = 5$

independent replicates). **e** Comparison of activation times for printed lines at different activation voltages. **f** Schematic illustration of the voltage-induced capillary effect of EGaIn droplets in LM-HIPEG. **g** Photos showing the voltage-induced capillary effect of a large EGaIn droplet in Carbopol hydrogel.

Moreover, EGaIn undergoes expansion during the solidification[50,51], so EGaIn and oxide shell have significant difference expansion coefficients at the melting point of EGaIn. As a result, when the temperature drops below the melting temperature of EGaIn, the stress caused by the volume expansion of EGaIn can lead to the rupture of the oxide shells. This rapture creates a connection between the EGaIn droplets, enabling electrical conductivity[18,20] (Fig. 4a, right part, and Fig. 4.d). The results of low-temperature solidification activation are depicted in Fig. 4e. At −29.2 °C, the conductivity of the printed line (210 μm) increased from 0.04 S m⁻¹ to $5.0 \times 10^5$ S m⁻¹. SEM image reveals the formation of a complete conductive pathway inside the line (Supplementary Fig. 21). By utilizing cryogenic solidification to activate the ink, high conductivity can be achieved without altering the macroscopic morphology of the printed samples. The original shape of the printed samples is well maintained when the LM transitions from solid to liquid state again (Supplementary Fig. 22, Supplementary Movie 3). This is due to a partial rupture of the oxide shells, while the remaining part can still preserve the shape of the EGaIn droplets, ensuring the shape stability of the macroscopic structure. This remarkable conductivity activation capability significantly broadens the potential applications of LM-HIPEG.

## Voltage-induced conductivity activation of LM-HIPEG by electrocapillary effect

We have also discovered that LM-HIPEG can achieve high conductivity through electric field effects. As shown in Fig. 5a, the as-printed line ($d = 210$ μm, $L = 3$ cm) exhibits a high resistance of $0.8 \times 10^6$ Ω within the bias range of 0 to 9 V (Fig. 5d). However, when the voltage across the line surpasses 9.3 V, the current rapidly increases. After 5 s, as the voltage increases to 14 V, the resistance drops to 1.2 Ω, enabling the printed line to become conductive. Figure 5b demonstrates the LED light is illuminated when it is connected to a printed line that is activated by a 20 V voltage. SEM image (Fig. 5c) of the activated line reveals that the droplets in the line no longer maintain their original polyhedral shape. Instead, adjacent droplets have coalesced, creating conductive pathways within the filament. Moreover, the surface of the line exhibits a wrinkled morphology, similar to that of the frozen LM ink (Supplementary Fig. 22). This morphology characteristic indicates partial rupture of the oxide thin layer on the droplets under the influence of the electric field. Figure 5d demonstrates that with the increase of the length of the printed ink line between the electrodes, a higher activation voltage is required, indicating a possible electric field-driven process. Additionally, for samples of equal length, the

activation time gradually increases as the voltage decreases. As shown in Fig. 5e, a 3.0 cm printed line achieves high conductivity within only 0.58 s at 15 V, compared to 15.40 s at 10 V. Unlike the partial merging of droplets in ink caused by thermal expansion, this method enables rapid and complete conductivity activation of LM-HIPEG ink under ambient temperature without mechanical stimulation (Supplementary Fig. 23). Unlike traditional strain-induced activation methods, such as pressing and stretching, which require a flexible matrix, this voltage-induced activation method can be employed in a rigid matrix, making it highly suitable for multi-material printing integration.

The voltage-induced electric conductivity activation described above is attributed to the electrocapillary effect[52,53]. The continuous phase of LM-HIPEG, Carbopol hydrogel, is essentially an electrolyte hydrogel. Thus, at the interface between the EGaIn droplets and the electrolyte gel in the ink, an electric double layer (EDL) is formed, as shown in Fig. 5f. This phenomenon is caused by charge transfer across the interface, adsorption of polymer chains or ions, or other processes. Upon applying an electric field to the ink, an electrocapillary phenomenon takes place in the EDL. The charges in the EDL are influenced by the electric field, causing ions in the diffusion layer of the continuous phase and charges in the EGaIn droplet to redistribution and consequently form a charge density gradient along the direction of the electric field. The redistribution of charges leads to a change in the interfacial tension. Since a charge density gradient is formed on the surface of the droplet, this also creates a gradient in interfacial tension on the droplet surface. To minimize the system's Gibbs energy, the LM flows from the region of low interfacial tension to that of high interfacial tension. This flow changes the overall shape of the LM droplet, causing the brittle oxide layer to rupture. The shape change can be observed directly on a larger droplet. As shown in Fig. 5g and Supplementary Movie 4, when a 20 V voltage was applied across the Carbopol hydrogel, a suspended droplet of EGaIn within the hydrogel underwent a sudden change in shape from ellipsoidal to spherical. This change in droplet shape caused the rupture of the oxide film on the surface, because it can be clearly observed that a color change of the droplet from gray to metallic luster was accompanied by the shape change. Since the electrode does not make direct contact with the LM droplet, this process is an electric field-induced electrocapillary phenomenon. Within the printed ink, the rupture of oxide films on adjacent droplets leads to their coalescence, leading to the formation of conductive pathways.

### The application of LM-HIPEG 3D printing

To demonstrate the application of LM-HIPEG in flexible electronic device, we conducted a printing of LM-HIPEG on a PDMS substrate to create a flexible and stretchable circuit (Fig. 6a-c). The conductivity of the printed wire remained satisfactory when the substrate was stretched and shrank, allowing the LED lights connected to the circuit to function properly. This successfully showcased the impressive stretchability of the printed circuit. In addition to meeting the requirements of printing on planar substrate, LM-HIPEG also allows for in-situ 3D printing on non-planar surfaces. In-situ printing on non-planar substate has become an important application of 3D printing technology[54–57]. Figure 6d−f demonstrates the successful printing of a 3D conductive circuit using LM-HIPEG on a PLA pyramid-like substrate. The pyramid-like substrate is designed with surface tilt angles of 0°, 45°, and 90°, and LM-HIPEG lines with a diameter of 210 µm are printed on it successfully. After the circuit was activated under a 20 V electric field, the nine LED lights connected to this circuit worked properly. This indicates that LM-HIPEG has the capability to be used for in-situ printing on the substrate with a tilt angle range of 0 ~ 90°, thanks to its 3D shaping ability and excellent interface compatibility with the printing surface.

Another unique feature of LM-HIPEG is its compatibility with alternate 3D printing with other inks, which enables multi-material printing for integrated manufacturing. Multi-material printing allows for the arrangement of two or more materials with distinct properties or functionalities in 3D space, significantly enhancing the design and manufacturing capabilities of 3D printing[11,58,59]. In our experiment, we demonstrated the capability of multi-material alternating printing using LM-HIPEG and polydimethylsiloxane (PDMS). As shown in Fig. 6 g, h, we utilized PDMS /polytetrafluoroethylene (PDMS/PTFE) ink[60] to 3D print an insulating matrix. Within the matrix, we embedded a pre-designed multi-layered LM-HIPEG circuit pattern with a predetermined interconnection between different layers, such as terminal I to III and II to IV (Supplementary Fig. 24). The printing process of this circuit is demonstrated in Supplementary Movie 5. Figure 6h shows that the printed multi-layered flexible circuit worked well to light two LED's. This printing technique overcomes the limitations of traditional conductive inks, which are restricted to 2D patterns, and is of significant importance for the in-situ construction of multi-layered 3D circuits. Additionally, we also achieved alternating printing of LM-HIPEG with high-strength thermosetting ink, epoxy resin/graphene nanoplatelet (EP/GN)[61], producing a printed object with a checkerboard pattern, as depicted in Fig. 6i. Detailed magnified images (Supplementary Fig. 25) reveal a strong interface bond between LM-HIPEG and the thermosetting ink, ensuring both structural stability of the printed materials and the functionality of individual ink materials. These experimental findings highlight the excellent printing compatibility of LM-HIPEG with various structurally and functionally suitable inks for DIW, enabling alternating 3D pattern printing. Such compatibility shows great potential for advancing the integration of 3D printing in flexible electronic device manufacturing.

## Discussion

In this work, we developed a 3D printable high internal phase emulsion gel LM ink. By exploiting the interaction between surface gallium oxides and Carbopol molecules, LM droplets were dispersed into Carbopol hydrogel in a high volume fraction of 82.5%, forming a high internal phase emulsion ink. The ink exhibits excellent elastic behavior (storage modulus ~ $10^4$ Pa) and shear-thinning rheological properties, which are attributed to the interfacial activity and lubricating effects of Carbopol on the surface of LM droplets. The ink can be used in direct ink-writing 3D printing, enabling the creation of 3D structures with a high resolution of 210 µm. This achievement marks the successful implementation of LM extrusion-based 3D printing for vertical stacked 3D objects. In addition, we developed an innovative voltage-induced electric conductivity activation method that exploits the electrical capillary behavior of LMs within electrolyte hydrogels. This method enables the achievement to achieve conductivity in LM emulsion-type inks by applying an electric field, without the need for external mechanical or thermal procedures. Based on the ink's excellent printability, LM conductive lines are 3D printed on flexible substrates and co-printed with multiple materials. In particular, the 3D printing of LM conductive wires on a vertical substrate was achieved, which expands the application of LM in complex structured devices. In summary, the developed LM 3D printing ink has the capability of high-resolution 3D printing and multi-material integration, which offers broad application prospects in fields such as flexible electronics and printing electronics.

## Methods
### Material

All chemical reagents were purchased from commercial sources without further purification. EGaIn LM (density: 6.94 g cm$^{-3}$) was provided by Shenyang Jiabei Trading Co., Ltd. Carbopol U20 was purchased from Lubrizol Corporation, USA. TEA-99 triethanolamine was provided by Shandong Youso Chemical Technology Co., Ltd.

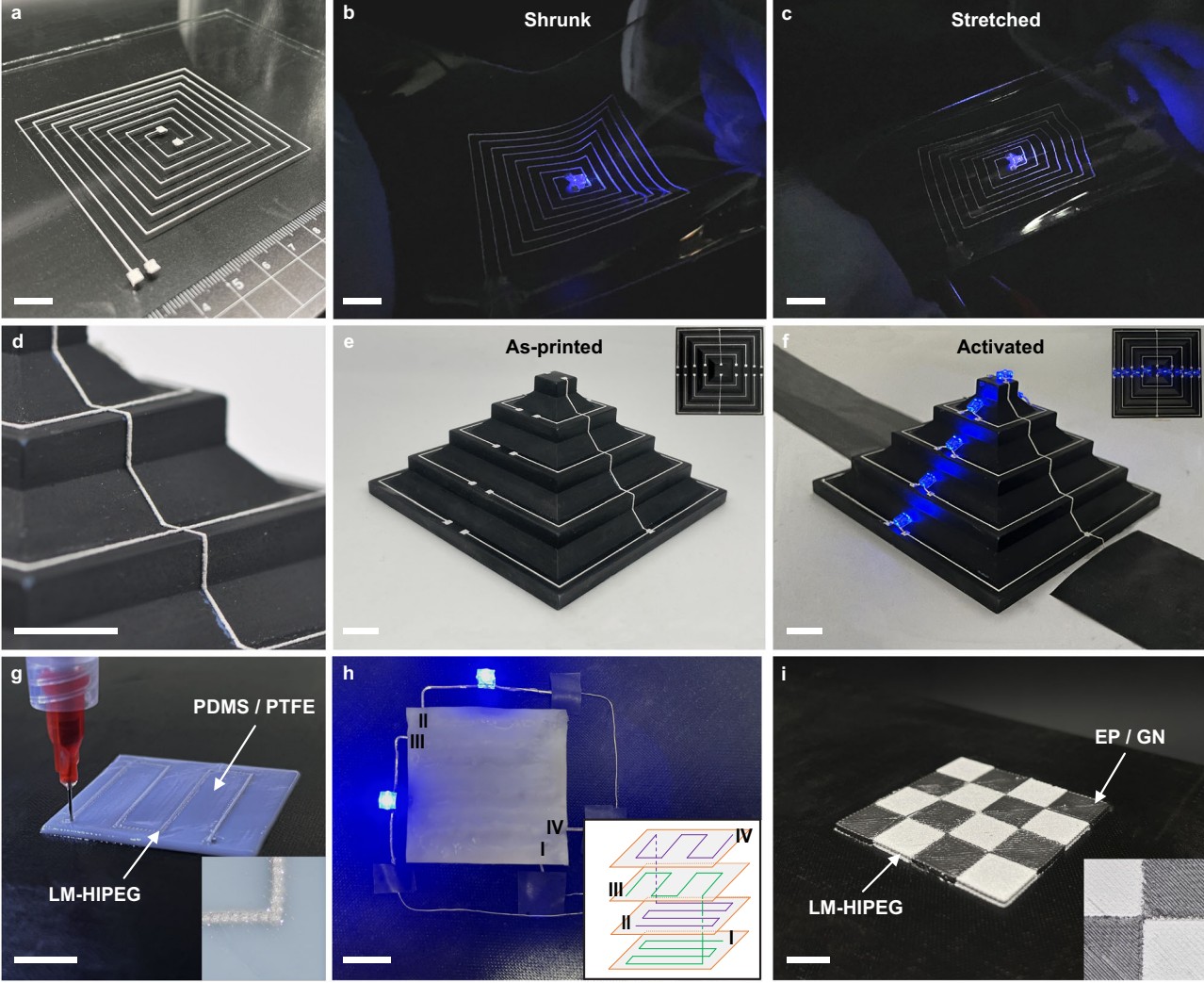

**Fig. 6 | The Application of LM-HIPEG Ink in 3D Printing. a** Photo of LM-HIPEG circuit on a flexible PDMS substrate. **b** - **c** Photos of the printed circuit working under shrinking (**b**) or stretching (**b**) of the PDMS substrate. **d** - **e** Photos of a 3D LM-HIPEG circuit in situ printed on a step-like substrate. **f** Photo showing the 3D circuit in (**e**) lighting 5 LEDs. **g** Alternative printing process of Multi-layered circuit composed of LM-HIPEG lines and PDMS/PTFE substrate. Inset is the magnification of the boundary part of the two materials. **h** Photo showing the 3D circuit in (**e**) lighting 5 LEDs. Inset shows the structure of the circuit. **i** Checkerboard-patterned object prepared by alternative printing of LM-HIPEG and EP/GN. Inset is the magnification of the boundary part of the two materials. (all scale bars in this figure are 1 cm).

Polydimethylsiloxane (PDMS 184 and Sylgard 527) was purchased from Dow Corning Corporation, USA. Bisphenol A epoxy resin, MeHHPA, and DMP-30 were purchased from Changzhou Runxiang Chemical Co., Ltd. Graphene nanoplatelet was supplied by Xiamen KNano Company. Polytetrafluoroethylene power was the product of Shanghai Macklin Biochemical Co., Ltd.

### Preparation of LM-HIPEG ink

240 mg of Carbopol was added to 36 g of deionized water and allowed to stand at room temperature for 24 h. Then, 275 mg of triethanolamine was added to adjust the pH to 7.0, resulting in the formation of Carbopol gel. 1.7 g of Carbopol gel was mixed with 50 g of EGaln in a beaker and stirred using a handheld shear emulsifier (Shanghai Huxi HR-6) at a speed of 25,000 rpm for 5 min. The mixture was then degassed at 25 °C in a vacuum oven for 15 min, resulting in the formation of LM-HIPEG ink, which was stored under a $N_2$ atmosphere.

### 3D printing of LM-HIPEG

The syringe containing LM-HIPEG ink was installed on a homemade ink extrusion-based 3D printer equipped with a pressure-driven extrusion system. The extrusion speed of the ink ranges from 5 t 35 mm s$^{-1}$ under air pressure between 370 and 430 kPa, the shear stress ($\tau$) ranges between 1417.5 - 1942.5 Pa, calculated using $\tau = -\Delta PR/2L$ [62] (Supplementary movie. 6). Nozzles with an inner diameter of 200 - 300 μm were used. A 3D geometric model in STL format was created, and then sliced to generate G-code files for the 3D printer. The layer height and gap were set to the same value as the diameter of the printing nozzle. For most of the time, both the extrusion speed and the printing speed were maintained at 30 mm s$^{-1}$.

### Non-planar in-situ printing of LM-HIPEG

The step-like non-planar substrate was printed using an Objet 30 Prime (Stratasys, Israel) stereolithography 3D printer. LM-HIPEG ink was printed onto the surface of a step-shaped substrate following the designed path using a 200 μm inner diameter nozzle.

### Multi-material printing of LM-HIPEG

A PDMS/PTFE ink was prepared with a mass ratio of Sylgard 527 curing agent to precursor to PTFE powder as 1:3:5 [60]. An EP pre-polymer was prepared by mixing bisphenol-A epoxy resin (E51), curing agent methyl

hexahydrophthalic anhydride (MeHHPA), and curing promoter tri(dimethylaminomethyl)phenol (DMP-30) in a ratio of 100:87.5:0.3. The EP/GN ink was prepared with a mass ratio of EP pre-polymer to GN as 10:1[61]. Each ink was loaded into the syringe, and the inks were alternately printed according to the designed path. All inks were printed using a nozzle with a diameter of 300 μm.

## Infrared spectroscopy

Infrared Spectroscopy. The infrared spectra of all samples were collected using the Nicolet IS 10 infrared spectrometer from Thermo-Fisher. In the experiment, liquid metal was mixed with Carbopol gel in a 1:1 volume ratio, and the same stirring program as used in the previous experiment with preparation of 82.5% LM-HIPEG was applied to achieve uniform dispersion of EGaIn. The mixture was then drop-cast onto KBr sample plates and subjected to 128 scans at a resolution of 2 cm$^{-1}$ in transmission mode to collect the infrared spectra. Additionally, after freeze-drying 0.067% Carbopol gel, its spectrum was collected using the ATR mode with a Ge crystal; the spectrum of the liquid metal was also collected in ATR mode.

## Rheological characteristics

The rheological properties of all samples were measured using an MCR 302 rotational rheometer (Anton Paar, Graz, Austria) with parallel plates with a diameter of 25 mm and a plate-plate gap of 0.5 mm.

## Transmission electron microscope

The prepared LM-HIPEG was slowly added into anhydrous ethanol for dispersion, A small amount of the dispersion was then placed onto a 200-mesh copper grid. After drying, the Talos F200 high-resolution scanning transmission electron microscope (HRSTEM) was utilized for imaging. For elemental analysis, a line scan was conducted using an EDS detector in the high-angle annular dark field (HAADF) mode.

## Characterization

The morphology and elemental distribution of the samples were observed using a SU-70 scanning electron microscope (Hitachi, Japan). X-ray photoelectron spectroscopy (XPS) spectra were recorded using a Quantum 2000 XPS system (Physical Electronics, USA). The particle size of the LM was measured using a laser particle size analyzer LS-POP (OMEC, Zhuhai, China). AFM measurements were conducted under environmental conditions using a Bruker Innova AFM. A silicon probe, having a spring constant of 0.4 N m$^{-1}$, was utilized for testing force-distance curves. Thermal gravimetric analysis under a nitrogen atmosphere was performed using a STA 409EP simultaneous thermal analyzer (Netzsch, Germany) with a temperature range of 25 °C to 500 °C and a heating rate of 10 °C min$^{-1}$. The conductivity of the samples was measured using a CHZ-01RC precision resistance-capacitance tester (Hangzhou Lingzhi Technology Co., Ltd., China).

## Data availability

The data that support the findings of this study are available in figshare with the identifier (https://doi.org/10.6084/m9.figshare.25551876).

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

## Acknowledgements

H.B. acknowledges the financial support from the Natural Science Foundation of China (21975210, 22179115), and the Innovation Laboratory for Sciences and Technologies of Energy Materials of Fujian Province (IKKEM) (HRTP-[2022]–44).

## Author contributions

H.B. conceived the concept, supervised the research, and wrote the paper. X.L.H. supervised the research and analyzed the experimental data. Z.W.L. designed and performed the experiments, collected and analyzed the data, and wrote the paper. X.W.Q. and Z.Q.S.C. helped the experiments on morphology characterization. J.L.L. assisted in 3D printing. Y.N.Z., X.P.L. and J.M.Z. analyzed the experimental data. All authors discussed the results and commented on the manuscript.

## Competing interests

The authors declare no competing interests.
