## [Peer Review File · Nature Communications]

High Internal Phase Emulsions Gel Ink for Direct-Ink-Writing 3D Printing of Liquid MetalEditorial Note:

This manuscript has been previously reviewed at another journal that is not operating a transparent peer review scheme. This document only contains reviewer comments and rebuttal letters for versions considered at *Nature Communications*.

REVIEWER COMMENTS

Reviewer #2 (Remarks to the Author):

This work describes 3D printable liquid metal and electrical activation after printing. The major comments suggest that the authors need to clarify the mechanism of action for Carbopol as a stabilizer and viscosity modifier, provide more quantitative analysis regarding the stabilization mechanism, and improve the discussion of the electrical activation mechanism.

Major issues:

Clarify the Role of Carbopol: The authors need to experimentally or theoretically demonstrate whether the rheological properties of Carbopol, specifically the elastic modulus (G') or the presence of carboxyl groups, are crucial for the stabilization of the LM drops. This could involve comparing Carbopol with other polymers that have similar rheological properties to determine if the stabilization effect is unique to Carbopol.

Quantify Carbopol Thickness: The manuscript should include calculations or measurements of the thickness of the Carbopol layer between LM droplets. This information could help understand the stabilization and structure by correlating the thickness with the volume fraction of LM in the mixture.

Compute Laplace Stress: The authors are advised to compute the Laplace stress at the Plateau borders where the LM droplets meet and compare it with the yield stress of the material. This comparison could provide a quantitative insight into the static stability of the LM composite during printing.

Reassess the Mechanism Discussion: The paragraph beginning with "We further investigated the mechanism..." may need to be revised or shortened, especially if the immediate oxide formation is a well-known phenomenon that does not contribute significantly to the novelty of the study.

Clarify Coordination Bonds: There should be a more detailed discussion on the coordination chemistry between Ga^{3+} ions and carboxyl groups. The authors should specify how many COOH groups can bind to the oxide surface of the LM and how this affects the printing process.

Shear Stress During Printing: The manuscript should describe the magnitude of the shear stress applied to the LM composite during the 3D printing process, as this is crucial information for readers interested in the practical applications of the technology.

Consider Shear Banding: The authors should investigate the role of shear banding in the stability of the LM composite during printing. A thick Carbopol layer may create bands that do not transfer shear stress effectively, contributing to the stability during the printing process.

Refine Electrical Activation Mechanism: The discussion on electrical activation should be improved by considering electrokinetics, specifically electroosmosis, as a potential mechanism rather than electrocapillarity. The manuscript should also correct any inaccuracies in the representation of electric field lines on the metal surface.

Minor issue:

Clarification of Carbopol U20 Description: The term "crosslinked copolymer" should be clarified to avoid confusion with chemical crosslinking that would prevent flow under shear stress. The authors should ensure the text accurately reflects the physical properties of Carbopol U20.

Reviewer #3 (Remarks to the Author):

The authors present a reasonable explanation for the enhanced printability of 82.5% LM gel ink,

which can appear counterintuitive at first glance. It encourages readers to consider the lubrication effects among inclusions within a dispersion/ emulsion, and how that impacts the macroscale rheological behavior of the mixture.

Another outstanding aspect of this work is the use of electrocapillarity and solidification to achieve partial coalescence among LM inclusions within the gel, without disturbing the macroscale structure of the printed part. These are indeed more practical alternatives to approaches such as 'mechanical sintering', especially for delicate structures with complex geometry. Although partial coalescence on a droplet-to-droplet level is difficult to prove, the data and images in Figure 5 and SI Figure 14 look convincing.

This manuscript is well-written. The discussion is structured and coherent, technical concepts are expressed clearly, and the figures are easy to follow.

Questions:

- "Carbopol U20 is a crosslinked copolymer of acrylic acid and C10-C30 alkyl acrylate..." It appears that the gel matrix is already crosslinked prior to extrusion. Can printed parts be reloaded into syringes and extruded again? Do the ink's rheological properties (and LM particle size distribution) change before and after extrusion?
- The diameter of the nozzle is about an order of magnitude larger than that of majority of the LM droplets. However, there are still larger LM droplets (close to 100 um diameter), and the continuous phase is a gel. Did you observe irreversible material deposition along the inner walls of the nozzle and syringe? Could you comment on the ink's tendency to clog or jam during extrusion?
- The LM-HIPEG has high yield stress and resistance to flow (Figure 2). How much pressure is necessary to extrude the ink through 200-300 um nozzle using this homemade extruder? Is there an optimal pressure window (for this pneumatic-dispense system) that's high enough to cause the ink to flow but not exceed the LM inclusions' puncture resistance?
- Typically, an auger extruder helps to keep the ink well-mixed during extrusion, but that's not really an option for a LM-in-gel ink like this. Did you observe gradations of particle size distribution or LM concentration throughout the print?
- "...because LM is not permeable to water, water within the lines is trapped" Did you observe any signs of reactions between LM oxide and water within printed parts over time? E.g., changes in size and/ or surface texture, outgassing, etc.?
- "The printed objects can remain stable for up to 3 hours in ambient conditions." (SI Figure 11) If the Carbopol hydrogel adsorbed around the LM droplets is able to protect them from rupture and coalescence under high shear from extrusion, why isn't it sufficient to stabilize the LM droplets within the 3D printed parts indefinitely (if left undisturbed)?

Point-by-point Response to Reviewers' Comments

Reviewer 1 Comments:

The manuscript by Lin et al. presents a liquid metal high internal phase emulsion gel for 3D printing of liquid metal with high resolution. The 3D printing of Ga alloy-based liquid metals is rather challenging, and they were typically printed as the core within a core-sheal structure filament. Here, the authors used the new lubricating effect of Carbopol gel to produce a shear-thinning liquid ink that can be directly 3D printed. The printed high-resolution 3D models are quite impressive. They also found the novel electrocapillarity phenomenon in the inks, and used it to realize the high conductivity.

This discovery offers a new method for controlling the ink's conductivity beyond mechanical means. These findings are important for the 3D printing of liquid metal and relative fields such as flexible electronics. The manuscript can be accepted after minor revision.

Comment 1. *Is there any change in the rheological properties of the inks after extrusion or shearing? The authors should prove that the storage module of the ink can recover to the original value after shear thinning.*

Reply: Thank you for your valuable suggestion. The Three Interval Thixotropy Test (3ITT) is an effective measure for evaluating the stress response and recovery rate of material polymer networks over time as they undergo breakage. Supplementary Fig. 11 shows the 3ITT results for 82.5% LM-HIPEG, indicating that the ink can recover 75% of its structural strength within 15 seconds and more than 95% within 35 seconds. This demonstrates that the ink is capable of quickly regaining its original viscoelastic state after being subjected to a high shear rate. The ink's G' and G'' rapid and reversible response under shear stress ensures the successful 3D printing of LM-HIPEG. Furthermore, we conducted rheological and particle size distribution tests on the ink after extrusion through a 210 μm nozzle. The experimental results (Supplementary Fig. 14) indicate that the viscosity and modulus of the recovered ink exhibit a negligible increase, which may be due to a slight loss of water during the recovery process. The particle size distribution of the ink remained unchanged, suggesting that emulsion breaking did not occur as the ink passed through the nozzle during the printing process. These results demonstrate that our ink possesses excellent stability.

The related content is added in the manuscript:

“The results of the Three Interval Thixotropy Test (3ITT) test indicate that the ink’s modulus is capable of rapidly responding and recovering as it transitions from a high shear to a low shear stage (Supplementary Fig. 11).” (Page 10, line 178 ~ 180)

“The rheological properties and particle size distribution of the 82% LM-HIPEG ink after extrusion remained virtually unchanged, and there was no residue found in the nozzle after extrusion (Supplementary Fig. 14).” (Page 12, line 215 ~ 217)

Supplementary Fig. 11 and Fig. 14 are added into the Supporting Information:

*“**Supplementary Fig. 11** | The 3ITT curve of 82.5% LM-HIPEG. Structural restoration under low shear strain conditions (amplitude strain = 0.01%) for 20 s; structural deformation under high shear conditions (amplitude strain = 200%) for 60 s.*

Supplementary Fig. 11 shows the 3ITT results for 82.5% LM-HIPEG, indicating that the ink can recover 75% of its structural strength within 15s and more than 95% within 35s.” (Supporting information, Page 13)

“Supplementary Fig. 14 | Properties of the 82.5% LM-HIPEG ink after extrusion. a Particle size distribution of 82.5% LM-HIPEG in its original state, after the first recycling, and after the second recycling. b Viscosity as a function of shear rate for LM-HIPEG(original recycle 1st, and recycle 2nd). c Modulus as a function of shear stress for LM-HIPEG(original recycle 1st, and recycle 2nd).”
 (Supporting information, Page 16)

Comment 2. The authors demonstrate that liquid metal can be dispersed into Carbopol gel with a volume fraction larger than 85%. Can liquid metal be dispersed into elastomers, such as PDMS? These matrices are attractive in flexible electronics.

Reply: Thank you for your valuable comment. Through experimental verification, it has been found that liquid metal cannot be stably dispersed in PDMS at a high volume fraction of 82.5%. In addition, we have also compared the dispersion effects of liquid metal in several other common polymer matrices, with detailed analysis and discussion available in response to comment 6.

Comment 3. What is the range of printing speed.

Reply: Thank you for your valuable comment. The ink can be extruded through a 210 μm nozzle under a pressure of over 370 kPa, with an extrusion speed exceeding 4.2 mm s^{-1} . Theoretically, if the printing speed matches the extrusion speed, the ink should be able to print standard models accurately according to the preset program. However, in the actual printing process, there is an inevitable idle phase during which ink continues to be extruded. Therefore, if the extrusion speed is too high, excessive ink may

be extruded during the idle phase, thereby affecting the print quality. Our experiments have shown that setting the printing speed in the range of 5 mm/s to 35 mm/s yields superior printing results(Supplementary movie. 6).

Figure R1. The printing status of the ink at different speeds (from a screenshot in Supplementary movie 6).

The related content is revised in the manuscript:

“The extrusion speed of the ink ranges from 5 to 35 mm s⁻¹ under air pressure between 370 and 430 kPa, the shear stress (τ) ranges between 1417.5 ~ 1942.5 Pa, calculated using $\tau = -\Delta PR/2L$ (Supplementary movie. 6).” (Page 25, line 438 ~ 440)

“For most of the time, both the extrusion speed and the printing speed were maintained at 30 mm s⁻¹.” (Page 25, line 443 ~ 444)

Supplementary movie. 6 is added into the Supporting Information.

Comment 4. For the electrocapillarity phenomenon, does the ion strength of the matrix gel influence the activation of the conductivity?

Reply: Thank you for your valuable comment. We dispersed EGaIn into Carbopol hydrogels containing NaCl mass fractions of 0 to 0.5% and prepared the ink under identical conditions (adding ions with high concentrations would impact the rheology of the ink). Then, we printed 3 cm lines with 210 μ m in diameter. As shown in Figure R2 below, the average activation voltages were 9.3, 9.18, 9.09,

9.01, and 8.97 V, respectively. The decrease in activation voltage with increasing ion concentration was small. Theoretically, the electrocapillary effect should become more pronounced with an increase in ion concentration. However, the hydrogel's volume fraction in the ink is only 17.5%, and the diameter of the liquid metal droplets is approximately 35 microns. Therefore, changes in the droplet surface tension are not particularly sensitive to variations in the gel's ion concentration.

Figure R2. The impact of ion strength of the hydrogel on activation of the conductivity.

Comment 5. The entire paper lacks diversity in characterization methods and is not persuasive. For instance, TEM only characterizes the oxide thickness; however, TEM can also characterize the crystal structure of the mixture. FTIR does not label peaks, and many other characterization methods are not standardized.

Reply: Thank you for your valuable suggestion. As the dispersed droplets were too large to find an appropriate diffraction area for analysis, we supplemented high-resolution TEM images and energy dispersive spectrometer (EDS) mapping data. The calculation of fringe spacing corresponded to the (2,0,0) planes of Ga_2O_3 , thereby demonstrating the formation of surface oxides. We further proved the adsorption thickness of Carbopol molecules after drying (10 nm) and the thickness of the oxide layer (5 nm) using EDS mapping. These results also provided evidence for the role of carboxylate ions in stabilizing the droplets. We also retested the IR spectra and labeled the peaks. Additionally, we have provided a more detailed and standardized description of the main characterizations in the

Experimental Methods section.

The related content is revised in the manuscript:

“Fig. 1 Preparation process and formation principle of LM-HIPEG. a Schematic diagram of the preparation of LM-HIPEG. b Photos LM-HIPEG. c Conceptual illustration of the emulsion structure of LM-HIPEG. d SEM morphology of LM-HIPEG. e-f TEM/EDS of LM-HIPEG. g Infrared spectra of Carbopol hydrogel, LM-HIPEG, and EGaIn.” (Page 5, line 86 ~ 89)

“During dispersion in the matrix, EGaIn comes into contact with air, forming an oxidized surface layer²⁻⁴. As shown in Supplementary Fig. 3, high-resolution transmission electron microscopy (TEM) and energy dispersive spectrometer (EDS) revealed a layer of gallium oxide approximately 5 nm thick on the surface of EGaIn droplets dispersed in ethanol, which can reduce the surface tension of EGaIn to some extent^{5,6}, facilitating its dispersion in the matrix.” (Page 6, line 111 ~ 115)

“In the LM-HIPEG system, Carbopol molecules can form strong adsorptive interactions with the oxide layer, significantly improving this instability. During stirring, in addition to oxide layer formation, Ga metal can react with H₂O and oxygen O₂ to produce GaOOH, which, upon ionization, releases gallium ions Ga³⁺, Ga³⁺ can coordinate with carboxylate ions, resulting in the adsorption of Carbopol molecules. TEM-EDS line scans (Fig. e-g) show larger amount of C and O elements on the droplet surface compared to those of unmodified LM droplet (Supplementary Fig. 3), confirming the presence

of an adsorbed layer of Carbopol molecules. Based on the element distribution, the adsorbed layer is estimated to be about 10 nm thick.” (Page 7, line 121 ~ 128)

Infrared spectroscopic analysis was employed to further reveal the role of Carbopol hydrogel in the formation of LM-HIPEG (Fig. 1g). For the Carbopol hydrogel, the characteristic bands in the range of 1800 ~ 1650 cm^{-1} are ascribed to the stretching vibration of carboxyl group ($\nu\text{C}=\text{O}$). The asymmetric and symmetric stretching vibrations of COO^- appear at 1650 ~ 1500 cm^{-1} and 1430 ~ 1370 cm^{-1} , respectively. EGaIn is dispersed into Carbopol hydrogel through stirring, leading to coordination between carboxylate ions and Ga^{3+} released from GaOOH groups on the surface. Spectroscopic analysis reveals the complete absence of the carboxyl band ($\nu\text{C}=\text{O}$), alongside a blue-shift in the asymmetric stretching vibration band of carboxylate. These changes indicate that all the carboxyl groups of Carbopol chains are coordinated with Ga^{3+} ions. This phenomenon is consistent with existing research on Ga^{3+} interactions with carboxylate complexes, suggesting the formation of inner-sphere coordination between Ga^{3+} and the carboxyl groups in Carbopol^{7,8}. The interaction leads to strong adsorption of the hydrogel layer on the surface of EGaIn droplets, further reducing the surface energy and enabling the dispersion of EGaIn droplets in the Carbopol hydrogel. Additionally, the electrostatic repulsion between Carbopol molecular chains increases the solution's viscosity, forming a gel layer. This hydrogel layer acts as a viscoelastic protective layer for EGaIn, providing spatial hindrance and lubrication, making the dispersion more stable.” (Page 7, line 129 ~ Page 9, line 144)

“Infrared Spectroscopy. The infrared spectra of all samples were collected using the Nicolet IS 10 infrared spectrometer from ThermoFisher. In the experiment, liquid metal was mixed with Carbopol gel in a 1:1 volume ratio, and the same stirring program as used in the previous experiment with preparation of 82.5% LM-HIPEG was applied to achieve uniform dispersion of EGaIn. The mixture was then drop-cast onto KBr sample plates and subjected to 128 scans at a resolution of 2 cm^{-1} in transmission mode to collect the infrared spectra. Additionally, after freeze-drying 0.067% Carbopol gel, its spectrum was collected using the ATR mode with a Ge crystal; the spectrum of the liquid metal was also collected in ATR mode.

Rheological characteristics. The rheological properties of all samples were measured using an MCR 302 rotational rheometer (Anton Paar, Graz, Austria) with parallel plates with a diameter of 25 mm

and a plate-plate gap of 0.5 mm.

Transmission Electron Microscope. The prepared LM-HIPEG was slowly added into anhydrous ethanol for dispersion, A small amount of the dispersion was then placed onto a 200-mesh copper grid. After drying, the Talos F200 high-resolution scanning transmission electron microscope (HRSTEM) was utilized for imaging. For elemental analysis, a line scan was conducted using an EDS detector in the high-angle annular dark field (HAADF) mode.” (Page 26, line 455 ~ Page 28, line 477)

“AFM measurements were conducted under environmental conditions using a Bruker Innova AFM. A silicon probe, having a spring constant of 0.4 N m^{-1} , was utilized for testing force-distance curves.” (Page 28, line 482 ~ Page 27, line 470)

Supplementary Fig. 3 is added into the Supporting Information:

“**Supplementary Fig. 3** | **a-b** TEM images of liquid metal droplets dispersed in ethanol. **c** TEM/EDS line scan moves from the surface of the liquid metal droplet towards the center.

Supplementary Fig. 3 shows TEM images of liquid metal droplets dispersed in ethanol. Supplementary Fig. 3a shows the thickness of the oxide layer is 5 nm. By utilizing Digital Micrograph software to calculate the interplanar spacing as 0.2235 nm (Supplementary Fig. 3b), which corresponds to the (200) planes of Ga_2O_3 , the presence of surface gallium oxide is thereby demonstrated. Based on the density of the element distribution (Supplementary Fig. 3c), the oxide layer is estimated to be about 5 nm thick, which also confirms that the main component of the oxide layer is gallium oxide.”

(Supporting information, Page 5)

Comment 6. The advantages of mixing Carbopol hydrogel with liquid metal are not clearly delineated by the authors. It is also unclear whether other hydrogels can be used in place of Carbopol. Further clarification on the specific benefits of using Carbopol and whether alternative hydrogels are viable would enhance the understanding of the study.

Reply: Thank you for your valuable suggestion. To demonstrate the superiority of Carbopol hydrogel, we attempted to disperse liquid metal at a volume fraction of 82.5% into PDMS 184, PDMS 527, 2% PEO hydrogel, and 5% PVA solution. However, as Supplementary Fig. 16 shows, these materials either failed to form stable emulsions or experienced severe emulsion breaking during stirring/extrusion due to the lack of functional groups that interact with the oxide layer of the liquid metal droplets or the absence of a gel layer that acts as a lubricant during the rheological process. Therefore, the rheological property of the gel is not the only factor that influences the formation and stability of LM ink. For Carbopol, the carboxyl groups providing strong interaction with Ga and suitable viscosity providing effective lubrication make it an excellent matrix for dispersing LM droplets. It should also be noted that Carbopol may not be the only candidate for this purpose. We believe that by following the principles outlined in this work, more gels can be identified as suitable matrices.

The related content is added in the manuscript:

“At the same volume fraction of EGaIn, we compared composite inks made from liquid metal and several other polymeric matrix materials, such as polydimethylsiloxane (PDMS), polyethylene oxide (PEO) hydrogel, and polyvinyl alcohol (PVA) solution. As illustrated in Supplementary Fig. 16, these materials either failed to form stable emulsions or experienced severe emulsion breaking during stirring/extrusion, due to the lack of functional groups that interact with the oxide layer of the liquid metal droplets or the absence of a gel layer that acts as a lubricant during shearing.” (Page 12, line 225 ~ Page 13 line 231)

Supplementary Fig. 16 is added into the Supporting Information:

“Supplementary Fig. 16 | Compared to common polymer materials. a Viscosity as a function of shear rate of 2% PEO hydrogel, 5% PVA solution, PDMS 527, PDMS184. **b** Modulus as a function of shear stress of PEO hydrogel, PVA solution, PDMS 527, PDMS184. **c** Images of composite materials composed of EGaIn and common polymer materials.

EGaIn struggles to form uniform droplets within PDMS 184 at a high volume fraction of 82.5%, because of PDMS 184's high viscosity. In contrast, PDMS 527 has a much lower viscosity than PDMS 184 and can mimic the rheological behavior of Carbopol hydrogel by adding PTFE. However, we observed obvious demulsification when EGaIn is dispersed in PDMS 527 and subjected to continuous shear force. This issue arises because the droplet surfaces lack a lubricating layer, and the shear force breaks the oxide layer due to the movement of polydimethylsiloxane molecules. Additionally, the lack of carboxyl groups in PEO (polyethylene Oxide) and PVA (polyvinyl Alcohol) molecules limits their interaction with the oxide layer of the dispersed EGaIn. This interaction is much weaker than the interaction between carboxylate ions and Ga^{3+} . As a result, the dispersion of EGaIn in PEO hydrogels and PVA solutions is uneven, and some visible metallic lusters are found on the ink surface. The severe demulsification observed when extruded through a needle further proves the instability of the droplet dispersion.” (Supporting information, Page 18-19)

Reviewer 2 Comments:

This work describes 3D printable liquid metal and electrical activation after printing. The major

comments suggest that the authors need to clarify the mechanism of action for Carbopol as a stabilizer and viscosity modifier, provide more quantitative analysis regarding the stabilization mechanism, and improve the discussion of the electrical activation mechanism.

Comment 1. Clarify the Role of Carbopol: The authors need to experimentally or theoretically demonstrate whether the rheological properties of Carbopol, specifically the elastic modulus (G') or the presence of carboxyl groups, are crucial for the stabilization of the LM drops. This could involve comparing Carbopol with other polymers that have similar rheological properties to determine if the stabilization effect is unique to Carbopol.

Reply: Thank you for your valuable suggestion. According to your suggestion, we performed additional experiments and confirmed that G' of the gel matrix is not crucial to the rheological property of ink and the stability of LM droplet.

We used the salt effect to control the rheological properties of Carbopol hydrogel. The viscosity and elastic modulus of Carbopol hydrogel gradually decreased with the increase of sodium chloride concentration (Supplementary Fig. 8a-b), but we found that although Carbopol hydrogel lost a great deal of elastic modulus, the liquid metal could still be dispersed in it, and the modulus of the obtained ink increased slightly (Supplementary Fig. 8c-d).

To demonstrate the superiority of Carbopol hydrogel, we attempted to disperse liquid metal at a volume fraction of 82.5% into PDMS 184, PDMS 527, 2% PEO hydrogel, and 5% PVA solution. However, as Supplementary Fig. 16 shows, these materials either failed to form stable emulsions or experienced severe emulsion breaking during stirring/extrusion due to the lack of functional groups that interact with the oxide layer of the liquid metal droplets or the absence of a gel layer that acts as a lubricant during the rheological process. Therefore, the rheological property of the gel is not the only factor that influences the formation and stability of LM ink. For Carbopol, the carboxyl groups providing strong interaction with Ga and suitable viscosity providing effective lubrication make it an excellent matrix for dispersing LM droplets. It should also be noted that Carbopol may not be the only candidate for this purpose. We believe that by following the principles outlined in this work, more gels can be identified as suitable matrices.

The related content is added in the manuscript:

“The elasticity of LM-HIPEG arises from the fluid nature of the dispersed EGaIn droplets, which can store energy by the deformation under applied stress, and it does not have a direct correlation with the modulus of the continuous phase (Supplementary Fig. 8).” (Page 10, line 166 ~ 168)

“At the same volume fraction of EGaIn, we compared composite inks made from liquid metal and several other polymeric matrix materials, such as polydimethylsiloxane (PDMS), polyethylene oxide (PEO) hydrogel, and polyvinyl alcohol (PVA) solution. As illustrated in Supplementary Fig. 16, these materials either failed to form stable emulsions or experienced severe emulsion breaking during stirring/extrusion, due to the lack of functional groups that interact with the oxide layer of the liquid metal droplets or the absence of a gel layer that acts as a lubricant during shearing.” (Page 12, line 225 ~ Page 13 line 231)

Supplementary Fig. 8 and Fig. 16 are added into the Supporting Information:

“**Supplementary Fig. 8** | The impact of ion concentration on the rheological behavior of Carbopol hydrogels and LM-HIPEG. **a** Viscosity as a function of shear rate of Carbopol hydrogel with varying mass fractions of NaCl. **b** Modulus as a function of shear stress of Carbopol hydrogel with varying mass fractions of NaCl. **c** Viscosity as a function of shear rate of LM-HIPEG with varying mass fractions of NaCl. **d** Modulus as a function of shear stress of LM-HIPEG with varying mass fractions of NaCl.

Adding sodium chloride to Carbopol hydrogel will cause the viscosity and elastic modulus of Carbopol hydrogel to decrease (Supplementary Fig. 8a-b). However, The liquid metal can still be dispersed into it, and the modulus of the obtained ink increases slightly (Supplementary Fig. 8c-d), possibly because the addition of ions reduces the interfacial tension between the liquid metal and the gel, making the dispersed droplet smaller. This result clearly demonstrates that the rheological properties of the ink are not determined by the modulus of the gel matrix.” (Supporting information, Page 10)

“**Supplementary Fig. 16** | Compared to common polymer materials. **a** Viscosity as a function of shear rate of 2% PEO hydrogel, 5% PVA solution, PDMS 527, PDMS184. **b** Modulus as a function of shear stress of PEO hydrogel, PVA solution, PDMS 527, PDMS184. **c** Images of composite materials composed of EGaIn and common polymer materials.

EGaIn struggles to form uniform droplets within PDMS 184 at a high volume fraction of 82.5%, because of PDMS 184's high viscosity. In contrast, PDMS 527 has a much lower viscosity than PDMS 184 and can mimic the rheological behavior of Carbopol hydrogel by adding PTFE. However, we observed obvious demulsification when EGaIn is dispersed in PDMS 527 and subjected to continuous shear force. This issue arises because the droplet surfaces lack a lubricating layer, and the shear force breaks the oxide layer due to the movement of polydimethylsiloxane molecules. Additionally, the lack of carboxyl groups in PEO (polyethylene Oxide) and PVA (polyvinyl Alcohol) molecules limits their

interaction with the oxide layer of the dispersed EGaIn. This interaction is much weaker than the interaction between carboxylate ions and Ga^{3+} . As a result, the dispersion of EGaIn in PEO hydrogels and PVA solutions is uneven, and some visible metallic lusters are found on the ink surface. The severe demulsification observed when extruded through a needle further proves the instability of the droplet dispersion.” (Supporting information, Page 18-19)

Comment 2. Quantify Carbopol Thickness: The manuscript should include calculations or measurements of the thickness of the Carbopol layer between LM droplets. This information could help understand the stabilization and structure by correlating the thickness with the volume fraction of LM in the mixture.

Reply: Thank you for your valuable suggestion. In the LM-HIPEG, the EGaIn droplets are approximated as spheres, each coated with a uniform Carbopol hydrogel layer, as depicted in the schematic below. Based on the volume ratio of liquid metal to hydrogel and the particle size distribution graphs, the thickness of the Carbopol gel layers in 75%, 77.5%, 80%, 82.5%, and 85% LM-HIPEG can be calculated as 2.4, 2.1, 1.8, 1.6, and 1.3 μm , respectively.

The related content is added in the manuscript:

“This means that the adjacent LM droplets are separated by a thin layer of adsorbed hydrogel, with the calculated thickness of this gel layer ranging from 2.38 to 1.33 μm (Supplementary Fig. 13).” (Page 12, line 210 ~ 212)

Supplementary Fig. 13 is added into the Supporting Information:

“Supplementary Fig. 13 | Quantify Carbopol Thickness.

In the LM-HIPEG, the liquid metal droplets are approximated as spheres, each coated with a uniform Carbopol hydrogel layer, as depicted in the schematic below. Based on the volume ratio of liquid metal to hydrogel and the particle size distribution graphs, the thickness of the Carbopol hydrogel layers in

75%, 77.5%, 80%, 82.5%, and 85% LM-HIPEG can be calculated as 2.4, 2.1, 1.8, 1.6, and 1.3 μm , respectively. The calculation process is as follows:

$$\sum \frac{4}{3} \pi n_i \left(\frac{d_i}{2}\right)^3 : \sum \frac{4}{3} \pi n_i \left[\left(\frac{d_i}{2} + \delta\right)^3 - \left(\frac{d_i}{2}\right)^3\right] = V_{\text{EGaIn}} : V_{\text{Hydrogel}}$$

Here, n_i and d_i represent the frequency and particle size, respectively, in the particle size distribution graph. δ denotes the thickness of hydrogel, and V represents volume.” Supporting information, Page 15)

Comment 3. Compute Laplace Stress: The authors are advised to compute the Laplace stress at the Plateau borders where the LM droplets meet and compare it with the yield stress of the material. This comparison could provide a quantitative insight into the static stability of the LM composite during printing.

Reply: Thank you for your valuable suggestion. The Laplace pressure (ΔP), induced by surface tension, affects the stability of droplets within an emulsion, particularly when the droplets are small. It describes the difference in pressure inside and outside the droplet, a discrepancy that tends to cause small droplets to be absorbed by larger droplets, thereby affecting the stability of the emulsion. Using the pendant drop method, we measured the surface tension of EGaIn to be 0.196 N/m. Based on the equation $\Delta P = 2\gamma/r$, we calculated ΔP to be 22.4 kPa. From this, we fitted the relationship between the volume fraction of EGaIn in LM-HIPEG and the storage modulus (G'), yield stress (τ_y). As shown in Supplementary Fig. 10:

$$G' = 3.02 \times \varphi_{\text{eff}} (\varphi_{\text{eff}} - 1.51) \Delta P + 1.99 \times 10^6$$

$$\tau_y = 2.34 \times \varphi_{\text{eff}} (\varphi_{\text{eff}} - 1.43) \Delta P + 27034$$

The fitting results are both close to the conclusions of previous literature^{9,10}. This reflects the critical impact of the EGaIn volume fraction on the stability and rheological behavior of the emulsion.

The related content is added in the manuscript:

“Based on the test results from Fig 2a, the relationships between the EGaIn volume fraction and both the energy storage modulus and the yield stress can be fitted respectively (Supplementary Fig. 10) as: $G' \sim \varphi_{\text{eff}} (\varphi_{\text{eff}} - 1.51) \Delta P$, $\tau_y \sim \varphi_{\text{eff}} (\varphi_{\text{eff}} - 1.43) \Delta P$, where ΔP represents the Laplace stress ($\Delta P = 2\gamma/r$). The fitting results are both close to the conclusions of previous literature^{9,10}.” (Page 10, line 175 ~ 178)

Supplementary Fig. 10 is added into the Supporting Information:

“Supplementary Fig. 10 | The relationships between the EGaIn volume fraction and both the energy storage modulus and the yield stress. a Nonlinear fitting of the volume of EGaIn and the storage modulus (G') of the ink. b Nonlinear fitting graph of the volume of EGaIn and the yield stress (τ_y) of the ink.” (Supporting information, Page 12)

Comment 4. Reassess the Mechanism Discussion: The paragraph beginning with "We further investigated the mechanism..." may need to be revised or shortened, especially if the immediate oxide formation is a well-known phenomenon that does not contribute significantly to the novelty of the study.

Reply: Thank you for your valuable suggestion. The content was revised by shortening the description of the oxide layer formation and adding discussion on the adsorption of Carbopol on the oxide layer. We supplemented high-resolution TEM images and energy dispersive spectrometer (EDS) mapping data. We further proved the adsorption thickness of Carbopol molecules after drying (10 nm) and the thickness of the oxide layer (5 nm) using EDS mapping. These results also provided evidence for the role of carboxyl groups in stabilizing the droplets.

The related content is revised in the manuscript:

“Fig. 1 Preparation process and formation principle of LM-HIPEG. a Schematic diagram of the preparation of LM-HIPEG. b Photos LM-HIPEG. c Conceptual illustration of the emulsion structure of LM-HIPEG. d SEM morphology of LM-HIPEG. e-f TEM/EDS of LM-HIPEG. g Infrared spectra of Carbopol hydrogel, LM-HIPEG, and EGaIn.” (Page 5, line 87 ~ 89)

“During dispersion in the matrix, EGaIn comes into contact with air, forming an oxidized surface layer²⁻⁴. As shown in Supplementary Fig. 3, high-resolution transmission electron microscopy (TEM) and energy dispersive spectrometer (EDS) revealed a layer of gallium oxide approximately 5 nm thick on the surface of EGaIn droplets dispersed in ethanol, which can reduce the surface tension of EGaIn to some extent⁵, facilitating its dispersion in the matrix. The change in the oxidation state of Ga before and after the formation of LM-HIPEG was analyzed by X-ray photoelectron spectroscopy (XPS) (Supplementary Fig. 4), indicating that a significant amount of oxide layer is generated when EGaIn is dispersed in the Carbopol hydrogel. However, this oxide layer is quite unstable due to its thinness and brittleness. For example, shear dispersion of EGaIn in pure water does indeed result in emulsion formation, but the dispersed EGaIn droplets easily aggregate, leading to demulsification (Supplementary Fig. 5). In the LM-HIPEG system, Carbopol molecules can form strong adsorptive interactions with the oxide layer, significantly improving this instability. During stirring, in addition to oxide layer formation, Ga metal can react with H₂O and oxygen O₂ to produce GaOOH, which,

upon ionization, releases gallium ions Ga^{3+} , Ga^{3+} can coordinate with carboxylate ions, resulting in the adsorption of Carbopol molecules. TEM-EDS line scans (Fig. e-g) show larger amount of C and O elements on the droplet surface compared to those of unmodified LM droplet (Supplementary Fig. 3), confirming the presence of an adsorbed layer of Carbopol molecules. Based on the element distribution, the adsorbed layer is estimated to be about 10 nm thick.” (Page 7, line 118 ~ Page 8, line 135)

Supplementary Fig. 3 and Fig. 4 are added into the Supporting Information:

“**Supplementary Fig. 3** | a-b TEM images of liquid metal droplets dispersed in ethanol. c TEM/EDS line scan moves from the surface of the liquid metal droplet towards the center.

Supplementary Fig. 3 shows TEM images of liquid metal droplets dispersed in ethanol. Supplementary Fig. 3a shows the thickness of the oxide layer is 5 nm. By utilizing Digital Micrograph software to calculate the interplanar spacing as 0.2235 nm (Supplementary Fig. 3b), which corresponds to the (200) planes of Ga_2O_3 , the presence of surface gallium oxide is thereby demonstrated. Based on the density of the element distribution (Supplementary Fig. 3c), the oxide layer is estimated to be about 5 nm thick, which also confirms that the main component of the oxide layer is gallium oxide.” (Supporting information, Page 5)

“Supplementary Fig. 4 | XPS spectra of LM-HIPEG.

This layer is believed to be composed of amorphous oxide. X-ray Photoelectron Spectroscopy (XPS) analysis (Supplementary Fig. 4) reveals changes in the oxidation states of Ga before and after the formation of LM-HIPEG. In the pure LM, Ga3/2p exhibits two distinct peaks corresponding to Ga³⁺/Ga²⁺ states (binding energy = 1116.5 eV) and a Ga⁰ state (binding energy = 1114.4 eV). The Ga3d line (overlapping with In4d) also represents Ga³⁺/Ga²⁺ states (binding energy = 18.9 eV) and a Ga⁰ state (binding energy = 16.7 eV). Following the formation of LM-HIPEG, there is an increase in the area ratio of peaks associated with the Ga³⁺/Ga²⁺ states in both Ga4d and Ga3/2p lines, indicating that there is a higher degree of Ga oxidation occurring. In comparison to gallium, indium is less prone to oxidation, and the In4d line in the spectrum does not show a significant change, suggesting that the surface layer on EGaIn droplets is primarily composed of gallium oxide.” (Supporting information, Page 5)

Comment 5. Clarify Coordination Bonds: There should be a more detailed discussion on the coordination chemistry between Ga³⁺ ions and carboxyl groups. The authors should specify how many COOH groups can bind to the oxide surface of the LM and how this affects the printing process.

Reply: Thank you for your valuable suggestion. Because Carbopol hydrogel only occupies less than 25% of the volume of the ink, and the mass fraction of Carbopol in the hydrogel is only 0.67%, the number of molecules adsorbed on the droplet surface is very low. This makes it difficult to directly investigate the bond between Ga³⁺ and carboxy by using reflective infrared or other spectroscopy in a wet state. Therefore, we prepared an ink with high Carbopol hydrogel content (50% in volume) and used IR spectroscopy of transmission mode to investigate it. This allowed us to observe some

characteristic peaks in the IR spectrum. The IR spectra showed the blue-shift of the COO^- peak and the complete disappearance of the carboxyl peak, indicating that the carboxyl groups were fully combined with the Ga ions to form carboxylates. Hence, in LM-HIPEG with fewer Carbopol molecules, it is believed that all the carboxyl groups coordinate with Ga^{3+} ions.

The interaction between carboxyl groups and Ga^{3+} ions is essential to the printability of the ink. It promotes the formation of a gel layer around the exterior of EGaIn droplets, significantly increasing the resistance of the droplets to fusion during the printing process. This layer also provides effective lubrication, allowing the ink to be smoothly extruded. As a comparison, PEO hydrogel and PVA solution (as described in comment 1), which exhibit weaker interactions with the surface of EGaIn droplets, cannot prevent the droplets from emerging. Severe demulsification was observed EGaIn/PEO and EGaIn/PVA inks were extruded through a needle.

The related content is added in the manuscript:

“Fig. 1 Preparation process and formation principle of LM-HIPEG. a Schematic diagram of the preparation of LM-HIPEG. b Photos LM-HIPEG. c Conceptual illustration of the emulsion structure of LM-HIPEG. d SEM morphology of LM-HIPEG. e-f TEM/EDS of LM-HIPEG. g Infrared spectra of Carbopol hydrogel, LM-HIPEG, and EGaIn.” (Page 5, line 85 ~ 89)

“Infrared spectroscopic analysis was employed to further reveal the role of Carbopol hydrogel in the

formation of LM-HIPEG (Fig. 1g). For the Carbopol hydrogel, the characteristic bands in the range of $1800 \sim 1650 \text{ cm}^{-1}$ are ascribed to the stretching vibration of carboxyl group ($\nu\text{C=O}$). The asymmetric and symmetric stretching vibrations of COO^- appear at $1650 \sim 1500 \text{ cm}^{-1}$ and $1430 \sim 1370 \text{ cm}^{-1}$, respectively. EGaIn is dispersed into Carbopol hydrogel through stirring, leading to coordination between carboxylate ions and Ga^{3+} released from GaOOH groups on the surface. Spectroscopic analysis reveals the complete absence of the carboxyl band ($\nu\text{C=O}$), alongside a blue-shift in the asymmetric stretching vibration band of carboxylate. These changes indicate that all the carboxyl groups of Carbopol chains are coordinated with Ga^{3+} ions. This phenomenon is consistent with existing research on Ga^{3+} interactions with carboxylate complexes, suggesting the formation of inner-sphere coordination between Ga^{3+} and the carboxyl groups in Carbopol^{7,8,11}. The interaction leads to strong adsorption of the hydrogel layer on the surface of EGaIn droplets, further reducing the surface energy and enabling the dispersion of EGaIn droplets in the Carbopol hydrogel. Additionally, the electrostatic repulsion between Carbopol molecular chains increases the solution's viscosity, forming a gel layer. This hydrogel layer acts as a viscoelastic protective layer for EGaIn, providing spatial hindrance and lubrication, making the dispersion more stable.” (Page 7, line 129 ~ Page 8 line 144)

Comment 6. Shear Stress During Printing: The manuscript should describe the magnitude of the shear stress applied to the LM composite during the 3D printing process, as this is crucial information for readers interested in the practical applications of the technology.

Reply: Thank you for your valuable suggestion. The flow of ink in the print head can be considered as fully developed laminar flow in a pipe¹, where the shear stress can be calculated using the following equation:

$$\tau = -\frac{R}{2} \times \frac{\Delta P}{L}$$

Here, R is the radius of the pipe, ΔP is the pressure difference between the two ends of the pipe, and L is the length of the pipe under consideration. In our printing process, $R = 105 \text{ }\mu\text{m}$, $L = 10 \text{ mm}$, and $\Delta P = 270 \sim 330 \text{ kPa}$. The calculated value of τ is between $1417.5 \sim 1942.5 \text{ Pa}$.

The related content is added in the manuscript:

“The extrusion speed of the ink ranges from 5 to 35 mm s^{-1} under air pressure between 370 and 430

kPa, so the shear stress(τ) ranges between 1417.5 ~ 1942.5 Pa, calculated using $\tau = -\Delta PR/2L$ ¹ (Supplementary movie. 6).” (Page 25, line 438 ~ 440)

Comment 7. Consider Shear Banding: The authors should investigate the role of shear banding in the stability of the LM composite during printing. A thick Carbopol layer may create bands that do not transfer shear stress effectively, contributing to the stability during the printing process.

Reply: Thank you for your valuable suggestion. LM-HIPEG comprises numerous droplets, so it can be seen as a jammed system¹². Unlike normal emulsions, such thixotropic yield-stress materials often show shear localization when flowing, forming shear bands: parts of the material move while others don't. In thixotropic yield-stress materials, the following three types of shear bands may occur¹³:

1. **Shear localization due to stress heterogeneity:** For yield materials, if the stress is uneven, part of the sample may be below the yield stress, while another part may be above the yield stress, leading immediately to shear localization. P. Coussot proposed a theoretical model in his research¹⁴, assuming that the viscosity of the material is generated by the competition between the ordering of the particle interaction network and its reshaping under shear, thereby indicating that general structured fluids are strongly influenced by their thixotropic nature. As shown in Figure R3 a below, the model indicates that stable uniform flow is possible for shear stresses above a critical value; however, for lower stresses, the material may exhibit shear banding or static. In fact, we have also observed this phenomenon when testing the rheological behavior of LM-HIPEG by rotational rheometer MCR302 (Anton Paar, Austria) with a 25 mm diameter parallel plate with a 0.5 mm plate–plate gap (Figure R3 b), where the shear rate-shear stress curve of LM-HIPEG is close to the model when applying an apparent shear rate. Inks with a higher volume content of liquid metal have rheological curves that closely match the model due to their enhanced thixotropic properties. Meanwhile, the rheological curve of ink with a 75% volume content of liquid metal is closer to that of conventional shear-thinning materials because it behaves more like a typical emulsion.

Figure R3. a Typical flow curve of a thixotropic yielding-stress fluid predicted by the model compared with the usual representation of the flow curve for an ideal yield stress fluid¹⁴ **b** The flow curves of LM-HIPEG with different LM volume fractions.

2. **The existence of a critical shear rate:** P. C. F. Møller proposed that thixotropic materials possess an inherent critical shear rate, and they have shown that flow below this critical shear rate is unstable¹⁵, leading to shear band flow. G. Ovarlez proposed such a curve model¹⁶, as shown in Figure R4 a, where the flow curve takes the form of a curve truncated below a critical shear rate (γ_c) associated with critical shear stress(τ_c): For stress below τ_c , no steady flow can occur, and for stress larger than τ_c , the shear rate (γ) is necessarily larger than the critical shear(γ_c). We find that the flow curves of inks with a higher volume fraction of liquid metal are closer to this model, because they are not just yield materials, but also thixotropic yield materials. Furthermore, G. Ovarlez also noted that if the flow is forced in a small enclosed area, complex effects might be observed under all shear rate conditions. This is because in cases where the shear region is very small, h (the height or thickness of the region) tends to approach zero, eventually reaching values that are of the same order of magnitude as the size of the components comprising the material, here the LM droplets. This means that the observed behavior, especially the relationship between apparent shear rate and resulting shear stress, does not reflect the properties of the uniform material. Instead, it is more likely to reflect the behavior of the components. We gradually reduced the testing distance from 1000 μm to 210 μm , and from Figure R4 b, it can be seen that the critical shear rate of the ink indeed increases as the shear area decreases.

Figure R4. a Typical flow curve obtained for some pasty materials under imposed, apparent shear rate: The solid line corresponds to the rheological behavior of the material in steady-state homogeneous flow.(cited from¹⁶) **b** Shear stress-shear rate flow curve at different testing plate gaps.

3. **Wall slip:** All types of dispersed systems exhibit slip because unless the bulk structure is perfectly maintained on the wall and coupled to the wall, and unless the dispersed phase and the continuous phase move in complete tandem across all length scales, it largely depends on the thickness of the apparent slip layer¹⁷. This kind of slip may be disadvantageous when conducting rheological performance tests, but during the printing process, it is beneficial. The wall slip produced by the apparent slip layer (Carbopol gel layer) prevents direct contact between the oxide layer and the needle head, thereby avoiding emulsion breaking.

In summary, there is no doubt that shear bands exist in LM-HIPEG, and these shear bands must be distributed along the Carbopol gel layer, as it acts as the continuous phase. At shear rates below the critical level, thicker Carbopol gel layers between the droplets may form bands that cannot effectively transmit shear stress, leading to localized shearing. This localization of shear is beneficial for the stability of the droplets, as it weakens the transmission of shear stress between droplets, thus protecting the oxide layer from rupturing. In our practical printing process, when the ink is just squeezed out with the minimum air pressure (corresponding stress ~ 1417 Pa, close to the critical shear stress, 1100 Pa), shear bands are generated in the ink. However, when the ink was extruded under high air pressure (1942.2 Pa), exceeding the critical shear stress, a stable flow was achieved, thereby eliminating the first and second types of shear bands. Therefore, the presence of shear bands primarily influences the stages of ink processing that involve shear forces or shear rates below the critical thresholds. This includes ink transfer, stacking, and forming during the printing process. Of course, during the printing process, the wall slip-induced shear bands always exist and reduce the contact between the tube wall

and the oxide layer, preventing emulsification and clogging.

The related content is added in the manuscript:

“In addition, the Carbopol gel layer will serve as the apparent slip layer of the ink, directly interfacing with the plastic syringe and the stainless steel printing needle. A thicker Carbopol gel layer can create bands that do not transfer shear stress effectively, resulting in beneficial wall slip-type shear bands during the printing process, which avoid damage to the oxide layer.” (Page 13, line 231 ~ 235)

Comment 8. Refine Electrical Activation Mechanism: The discussion on electrical activation should be improved by considering electrokinetics, specifically electroosmosis, as a potential mechanism rather than electrocapillarity. The manuscript should also correct any inaccuracies in the representation of electric field lines on the metal surface.

Reply: Thank you for your valuable suggestion. Electroosmosis is the phenomenon that occurs when an electric field is applied to a fluid within a porous material or along a surface with a fixed electric charge. This electric field causes ions in the fluid to move, leading to a net movement of the fluid in the opposite direction of the mobile counterions in the electrical double layer due to interactions with the immobile charged surface. In LM-HIPEG, with the hydrogel layer as the continuous phase, electroosmosis may promote the movement of water through the interstitial gaps between droplets. However, this movement is not expected to result in the rupture of the oxide layer and subsequent merging of droplets.

Electrocapillarity, on the other hand, refers to the effect of an electric field on the surface tension of a liquid. It typically manifests as a change in the liquid surface tension at the interface between a solid electrode and the liquid when a voltage is applied. In LM-HIPEG, droplets usually exist in a polyhedral shape. If the surface tension of EGaIn changes due to electrocapillarity, it could change shape and potentially cause the rupture of the oxide layer.

In order to determine which effect dominates the activation of the ink, we examined the activation of the dried ink. A printed line of 40 mm in length was dried at 25°C and 40% air humidity to constant weight and submitted to conductivity activation by voltage. It was observed that the dried line could be activated at lower voltages compared with the original wet line (Figure R5). This result indicates

that electroosmosis is not a potential activation mechanism since the dried lines no longer contain water. But the electrocapillarity effect does not depend on the presence of water. In the dried printed lines, the induced electromotive force within the EGaIn droplets can cause a change in the charge density of the droplets' double layer, thereby altering the surface tension of droplets. Since there is no longer gel layer covering the droplets, the oxide layer is more easily ruptured. Therefore, we believe that the key to the electric activation mechanism lies in whether the surface tension of the EGaIn droplets changes. We have revised the electric field lines on the metal surface in Fig 5 of the manuscript.

Figure R5. Current-voltage curve of an as-printed line and a dried line.

The related figure is revised in the manuscript:

Comment 9. Clarification of Carbopol U20 Description: The term "crosslinked copolymer" should be clarified to avoid confusion with chemical crosslinking that would prevent flow under shear stress. The authors should ensure the text accurately reflects the physical properties of Carbopol U20.

Reply: Thank you for your valuable suggestion. Carbopol is a high molecular weight polymer primarily formed through the free radical polymerization of acrylic acid monomers. In this process, cross-linking agents, such as allyl ethers, are typically used to facilitate cross-linking reactions between acrylic acid monomers, resulting in a polymer with a three-dimensional network structure. However, a critical aspect of Carbopol's design is its relatively low degree of cross-linking. This design enables the polymer to swell and form a highly viscous gel upon hydration. The modest cross-linking level also avoids intermolecular bonding, endowing the gel with a certain degree of fluidity.

Specifically, the type of Carbopol utilized in our experiments is Ultrez 20, a white powder that is a hydrophobically modified, cross-linked acrylate copolymer. This physical property, along with other relevant details about Carbopol Ultrez 20, is documented on the manufacturer's official website (<https://www.lubrizol.com/Personal-Care/Products/Product-Finder/Products-Data/Carbopol-Ultrez-20-polymer>) and within the scientific literature.

The related content is revised in the manuscript:

“Carbopol U20 is a cross-linked copolymer of acrylic acid and C₁₀-C₃₀ alkyl acrylate, widely used as a rheology-modifying thickener^{18,19}.” (Page 5, line 94 ~ Page 6, line 95)

Reviewer 3 Comments:

The authors present a reasonable explanation for the enhanced printability of 82.5% LM gel ink, which can appear counterintuitive at first glance. It encourages readers to consider the lubrication effects among inclusions within a dispersion/ emulsion, and how that impacts the macroscale rheological behavior of the mixture.

Another outstanding aspect of this work is the use of electrocapillarity and solidification to achieve partial coalescence among LM inclusions within the gel, without disturbing the macroscale structure of the printed part. These are indeed more practical alternatives to approaches such as ‘mechanical sintering’, especially for delicate structures with complex geometry. Although partial coalescence on a droplet-to-droplet level is difficult to prove, the data and images in Figure 5 and SI Figure 14 look convincing.

This manuscript is well-written. The discussion is structured and coherent, technical concepts are expressed clearly, and the figures are easy to follow.

Comment 1. “Carbopol U20 is a crosslinked copolymer of acrylic acid and C10-C30 alkyl acrylate...”
It appears that the gel matrix is already crosslinked prior to extrusion. Can printed parts be reloaded into syringes and extruded again? Do the ink’s rheological properties (and LM particle size distribution) change before and after extrusion?

Reply: Thank you for your valuable suggestion. Carbopol is a high molecular weight polymer primarily formed through the free radical polymerization of acrylic acid monomers. In this process, cross-linking agents, such as allyl ethers, are typically used to facilitate cross-linking reactions between acrylic acid monomers, resulting in a polymer with a three-dimensional network structure. However, a critical aspect of Carbopol’s design is its relatively low degree of cross-linking. This design enables the polymer to swell and form a highly viscous gel upon hydration. The modest cross-linking level also avoid intermolecular bonding, endowing the gel with a certain degree of fluidity.

Specifically, the type of Carbopol utilized in our experiments is Ultrez 20, a white powder that is a hydrophobically-modified, cross-linked acrylate copolymer. This physical property, a

long with other relevant details about Carbopol Ultrez 20, is documented both on the manufacturer's official website (<https://www.lubrizol.com/Personal-Care/Products/Product-Finder/Products-Data/Carbopol-Ultrez-20-polymer>) and within the scientific literature^{18,19}.

We conducted rheological and particle size distribution tests on the ink extruded through a 210 μm nozzle. The experimental results indicate that the viscosity and modulus of the recovered ink exhibit a negligible increase, possibly due to a slight loss of water during the recovery process. The droplet size distribution of the ink remained unchanged, suggesting that emulsion breaking does not occur as the ink passes through the nozzle during the printing process. This demonstrates that our ink possesses excellent stability.

The related content is revised in the manuscript:

“Carbopol U20 is a cross-linked copolymer of acrylic acid and C₁₀-C₃₀ alkyl acrylate, widely used as a rheology-modifying thickener^{18,19}.” (Page 5, line 94 ~ Page 6, line 95)

“The rheological properties and particle size distribution of the 82% LM-HIPEG ink after extrusion remained virtually unchanged, and there was no residue found in the nozzle after extrusion (Supplementary Fig. 14).” (Page 12, line 215 ~ 217)

Supplementary Fig. 14 is added into the Supporting Information:

“Supplementary Fig. 14 | Properties of the 82.5% LM-HIPEG ink after extrusion. a Particle size distribution of 82.5% LM-HIPEG in its original state, after the first recycling, and after the second

*recycling. b Viscosity as a function of shear rate for LM-HIPEG(original recycle 1st, and recycle 2nd).
c Modulus as a function of shear stress for LM-HIPEG(original recycle 1st, and recycle 2nd).”*

(Supporting information, Page 16)

Comment 2. The diameter of the nozzle is about an order of magnitude larger than that of majority of the LM droplets. However, there are still larger LM droplets (close to 100 μm diameter), and the continuous phase is a gel. Did you observe irreversible material deposition along the inner walls of the nozzle and syringe? Could you comment on the ink’s tendency to clog or jam during extrusion?

Reply: Thank you for your valuable suggestion. As demonstrated by Figure R6 below, there is no residual ink observed on the walls of the plastic syringe. After printing, the stainless steel printing needle was then ultrasonically cleaned in anhydrous ethanol for 1 hour, and no residues are detected in the ethanol. Consequently, no residue is observed neither in the printing needle nor in the syringe. Therefore, our ink is quite stable during the printing process.

In fact, if the oxide layer rupture and contact the tube walls, it would adhere to them. Such adherence would lead to a continuous accumulation of the oxide layer on the walls, eventually causing blockages. However, a sufficiently thick Carbopol gel layer forms bands that do not efficiently transfer shear stress. These bands facilitate a type of wall slip during the printing process, which prevents the rupture of the oxide layer, thereby avoiding the adhesion and accumulation that lead to blockages.

Figure R6. The walls of the plastic syringe did not retain any ink after printing.

The related content is revised in the manuscript:

“In addition, the Carbopol gel layer will serve as the apparent slip layer of the ink, directly interfacing with the plastic syringe and the stainless steel printing needle. A thicker Carbopol gel layer can create bands that do not transfer shear stress effectively, resulting in beneficial wall slip-type shear bands during the printing process, which avoid damage to the oxide layer.” (Page 13, line 231 ~ 235)

Comment 3. The LM-HIPEG has high yield stress and resistance to flow (Figure 2). How much pressure is necessary to extrude the ink through 200-300 μm nozzle using this homemade extruder? Is there an optimal pressure window (for this pneumatic-dispense system) that’s high enough to cause the ink to flow but not exceed the LM inclusions’ puncture resistance?

Reply: Thank you for your valuable comment. The ink can be extruded through a 210 μm nozzle under a pressure of over 370 kPa, with an extrusion speed exceeding 4.2 mm s^{-1} . Theoretically, if the printing speed matches the extrusion speed, the ink should be able to print standard models accurately according to the preset program. However, in the actual printing process, there is an inevitable idle phase during which ink continues to be extruded. Therefore, if the extrusion speed is too high, excessive ink may be extruded during the idle phase, thereby affecting the print quality. Our experiments have shown that setting the printing speed in the range of 5 mm/s to 35 mm/s yields superior printing results(Supplementary movie. 6).

Figure R7. The printing status of the ink at different speeds (from a screenshot in Supplementary movie 6).

The related content is revised in the manuscript:

“The extrusion speed of the ink ranges from 5 to 35 mm s⁻¹ under air pressure between 370 and 430 kPa, the shear stress(τ) ranges between 1417.5 ~ 1942.5 Pa, calculated using $\tau = -\Delta PR/2L$ (Supplementary movie. 6).” (Page 25, line 438 ~ 440)

“For most of the time, both the extrusion speed and the printing speed were maintained at 30 mm s⁻¹.”
(Page 25, line 443 ~ 444)

Supplementary movie. 6 is added into the Supporting Information.

Comment 4. Typically, an auger extruder helps to keep the ink well-mixed during extrusion, but that’s not really an option for a LM-in-gel ink like this. Did you observe gradations of particle size distribution or LM concentration throughout the print?

Reply: Thank you for your valuable suggestion. We fully agree that the auger extruder is not suitable for our ink. The ink extrusion device we use, as shown in Figure R8, stores the pre-mixed ink in a syringe. The ink is directly extruded from a stainless steel needle using an air pump and a dispensing machine to control the air pressure, rather than a screw extruder. Because the ink is rather stable, the droplet size remains unchanged during printing. This is demonstrated by the consistent particle size distribution of the ink after extrusion, as described in the reply to comment 2.

Figure R8. The ink extrusion device.

Comment 5. "...because LM is not permeable to water, water within the lines is trapped" Did you observe any signs of reactions between LM oxide and water within printed parts over time? E.g., changes in size and/ or surface texture, outgassing, etc.?

Reply: Thank you for your valuable suggestion. It was reported that Ga can react with water, slowly releasing hydrogen. However, we did not observe any significant reaction during the printing process. One reason is that the initial water content in ink is as low as 1.67%, which decreased to 0.90% after being stored in air for 24 hours (Supplementary Fig. 17). Besides, the oxide layer covering the LM droplets can prevent the reaction between LM and water.

As indicated in SI Figure 11, the main visible changes in the printed object are attributed to the merging of droplets rather than chemical reactions. This merging is caused by the gravitational pull on the upper layer of material and the pressure exerted by the surrounding droplets, which, following the gradual evaporation of water, leads to the rupture of the oxide layer and continuous integration of the droplets.

Comment 6. "The printed objects can remain stable for up to 3 hours in ambient conditions." (SI Figure 11) If the Carbopol hydrogel adsorbed around the LM droplets is able to protect them from rupture and coalescence under high shear from extrusion, why isn't it sufficient to stabilize the LM droplets within the 3D printed parts indefinitely (if left undisturbed)?

Reply: Thank you for your valuable suggestion. The statement "LM is impermeable to water (Supplementary Fig. 10) might suggest that water within the printed lines is completely trapped" could lead to misunderstanding. Although the tightly packed liquid metal acts as an effective barrier, water evaporation through the gaps between droplets is inevitable. It would be more accurate to describe the process as one where the evaporation of water is slowed down. As the water gradually evaporates, the liquid metal droplets lose the protection of the gel layer. This, combined with the gravitational force of the upper layers of material and the pressure exerted by the surrounding droplets, leads to the rupture of the oxide layer. The continuous merging of droplets can potentially compromise the structural integrity.

The related content is added in the manuscript:

“However, because EGaIn is not permeable to water(Supplementary Fig. 17), the evaporation process of water within the ink is significantly slowed down, which means that after printing, the printed lines undergo minimal shrinkage and maintain their cylindrical shape.” (Page 15, line 251 ~ 254)

REFERENCES

1. Hardin, J. O., Ober, T. J., Valentine, A. D. & Lewis, J. A. Microfluidic Printheads for Multimaterial 3D Printing of Viscoelastic Inks. *Adv. Mater.* **27**, 3279-3284 (2015).
2. Regan, M. et al. X-ray study of the oxidation of liquid-gallium surfaces. *Phys. Rev. B* **55**, 10786 (1997).
3. Lin, Y., Genzer, J. & Dickey, M. D. Attributes, Fabrication, and Applications of Gallium-Based Liquid Metal Particles. *Adv. Sci.* **7**, 2000192 (2020).
4. Larsen, R. J., Dickey, M. D., Whitesides, G. M. & Weitz, D. A. Viscoelastic properties of oxide-coated liquid metals. *J. Rheol.* **53**, 1305-1326 (2009).
5. Kim, D. et al. Recovery of Nonwetting Characteristics by Surface Modification of Gallium-Based Liquid Metal Droplets Using Hydrochloric Acid Vapor. *ACS Appl. Mater. Interfaces* **5**, 179-185 (2013).
6. Liu, T., Sen, P. & Kim, C. J. Characterization of Nontoxic Liquid-Metal Alloy Galinstan for Applications in Microdevices. *J. Microelectromech. Syst.* **21**, 443-450 (2012).
7. Clausén, M., Öhman, L. O. & Persson, P. Spectroscopic studies of aqueous gallium(III) and aluminum(III) citrate complexes. *J. Inorg. Biochem.* **99**, 716-726 (2005).
8. Hafiz, S. S. et al. Surfaces and Interfaces of Liquid Metal Core-Shell Nanoparticles under the Microscope. *Part. Part. Syst. Charact.* **37**, 1900469 (2020).
9. Mason, T., Bibette, J. & Weitz, D. Elasticity of compressed emulsions. *Phys. Rev. Lett.* **75**, 2051 (1995).
10. Mason, T. G., Bibette, J. & Weitz, D. A. Yielding and flow of monodisperse emulsions. *J. Colloid Interface Sci.* **179**, 439-448 (1996).
11. Hagvall, K., Persson, P. & Karlsson, T. Spectroscopic characterization of the coordination chemistry and hydrolysis of gallium(III) in the presence of aquatic organic matter. *Geochim. Cosmochim. Acta* **146**, 76-89 (2014).
12. Goyon, J., Colin, A., Ovarlez, G., Ajdari, A. & Bocquet, L. Spatial cooperativity in soft glassy flows. *Nature* **454**, 84-87 (2008).
13. Paredes, J., Shahidzadeh-Bonn, N. & Bonn, D. Shear banding in thixotropic and normal emulsions. *J. Phys.-Condens. Mat.* **23**, 284116 (2011).

14. Coussot, P., Nguyen, Q. D., Huynh, H. T. & Bonn, D. Viscosity bifurcation in thixotropic, yielding fluids. *J. Rheol.* **46**, 573-589 (2002).
15. Moller, P., Fall, A., Chikkadi, V., Derks, D. & Bonn, D. An attempt to categorize yield stress fluid behaviour. *Philos. T. R. Soc. A.* **367**, 5139-5155 (2009).
16. Ovarlez, G., Rodts, S., Chateau, X. & Coussot, P. Phenomenology and physical origin of shear localization and shear banding in complex fluids. *Rheol. Acta* **48**, 831-844 (2009).
17. Bertola, V., Bertrand, F., Tabuteau, H., Bonn, D. & Coussot, P. Wall slip and yielding in pasty materials. *J. Rheol.* **47**, 1211-1226 (2003).
18. Dinkgreve, M., Fazilati, M., Denn, M. M. & Bonn, D. Carbopol: From a simple to a thixotropic yield stress. *J. Rheol.* **62**, 773-780 (2018).
19. Khan, N. H., Paswan, M. K. & Hassan, M. A. Thermorheological characterization and ANN modelling of aqueous carbopol-titania yield stress nanofluid. *Thermochim. Acta* **720**, 179418 (2023).

REVIEWERS' COMMENTS

Reviewer #2 (Remarks to the Author):

I am satisfied with most of the author's answers other than 8.

"Electrocapillarity, on the other hand, refers to the effect of an electric field on the surface tension of a liquid. It typically manifests as a change in the liquid surface tension at the interface between a solid electrode and the liquid when a voltage is applied. In LM-HIPEG, droplets usually exist in a polyhedral shape. If the surface tension of EGaIn changes due to electrocapillarity, it could change shape and potentially cause the rupture of the oxide layer. "

In the figure, it clearly shows the counterions, which makes it confusing. please explain step by step how electric field changes the interfacial tension in the presence/absence of counter ions if you want to argue that it is from the electrocapillary that I do not understand.

Once it is clarified, I think it can be published pretty much as it is.

Reviewer #3 (Remarks to the Author):

Thank you for addressing the prior comments, which I've summarized here:

Comment 1: Addressed.

Comment 2: Addressed.

Comment 3: Addressed.

Comment 4: Addressed.

Comment 5: Addressed – the reasoning provided makes sense. (Minor note: the oxide layer itself can react with water.)

Comment 6: Addressed. Follow-up question: Since gravitational forces pressurize LM particles in the 3D structure into coalescence, does that mean 2D traces exhibit better shape retention (longer than 3 hours)?

New questions/ suggestions:

Comment 7: Regarding voltage induced activation, the authors provide a good depiction of the electrocapillary force acting on a droplet in Fig 5g. However, is the activation caused by this voltage-induced motion exclusively? Could thermal expansion, via joule heating, cause a similar effect, and did the authors perform any experiments to eliminate that as a possible cause?

Comment 8: In the text, it is clear that the LM-HIPEG solidifies by freezing at -30 °C. This is not as clear in Fig 3h and would be helpful to have the caption mention this.

Comment 9: In Figure 2e, there are two y-axis, but it is not clear which one applies to which curve. Color coordination would helpful here.

Comment 10: The following papers may be of interest to the authors:

- "Ultrastretchable, Wearable Triboelectric Nanogenerator Based on Sedimented Liquid Metal Elastomer Composite" – Pan, et al. - <https://doi.org/10.1002/admt.202000754>. This paper uses density driven sedimentation to produce a locally concentrated liquid metal elastomer composite, which becomes conductive once the liquid metal particles become densely packed.
- "Directed Assembly of Liquid Metal-Elastomer Conductors for Stretchable and Self-Healing Electronics" – Krisnadi, et al. - <https://doi.org/10.1002/adma.202001642>. Although not a printable ink, this paper uses electric field to activate a liquid metal elastomer composite.
- "Deformable liquid metal polymer composites with tunable electronic and mechanical properties" – Koh, et al. - <https://doi.org/10.1557/jmr.2018.209>. This early work on LM-elastomer composites shows a conductive inverted phase emulsion, where galinstan is the continuous phase.

Point-by-point Response to Reviewers' Comments

Reviewer 1 Comments:

"Electrocapillarity, on the other hand, refers to the effect of an electric field on the surface tension of a liquid. It typically manifests as a change in the liquid surface tension at the interface between a solid electrode and the liquid when a voltage is applied. In LM-HIPEG, droplets usually exist in a polyhedral shape. If the surface tension of EGaIn changes due to electrocapillarity, it could change shape and potentially cause the rupture of the oxide layer. "

In the figure, it clearly shows the counterions, which makes it confusing. please explain step by step how electric field changes the interfacial tension in the presence/absence of counter ions if you want to argue that it is from the electrocapillary that I do not understand.

Reply: Thank you for your valuable comment. The counter ions are shown in the figure because they take part in the electrocapillary process. The continuous phase, Carnopol hydrogel, contains electrolytes, so an electric double layer (EDL)^{1,2} will form at the interface between liquid metal droplets and the hydrogel. This phenomenon is caused by charge transfer across the interface, adsorption of polymer chains or ions, or other processes. Therefore, there are charges near the surface of the EGaIn droplets and ion diffusion layer in the hydrogel. When an external electric field is applied, the charges in the EDL are influenced by the electric field, causing ions in the diffusion layer of the continuous phase and charges in the EGaIn droplet to rearrange and consequently form a charge density gradient along the direction of the electric field. It should be noted that there is repulsion between charges in the EDL, so the charge density at the interface contributes to the interfacial tension. High-density interface charges will make it easier to increase the interface area by providing repulsion, resulting in smaller interfacial tension. Therefore, in our case, the redistribution of charges leads to a change in the interfacial tension. Since a charge density gradient is formed on the surface of the droplet, this also creates a gradient in interfacial tension on the droplet surface, thereby affecting the overall shape of the liquid metal particles. This results in the rupture of the brittle oxide layer, allowing the liquid metal within the particles to escape and merge, forming conductive pathways.

The process has been described by the electrocapillary curve³. Figure R1 shows a typical

electrocapillary curve, from which we can find that the change in the electric potential across the metal/electrolyte interface will significantly change the interfacial tension. According to the electrocapillary theory, the concentration of the electrolyte can influence the electrocapillary effect. High-concentration electrolytes lead to a more pronounced electrocapillary effect due to the higher charge density in EDL.

Figure R1. Electrocapillary curve showing the interfacial tension of a liquid metal drop as a function of electric potential. The apex of the curve represents the potential of zero charge; any change in voltage, whether positively or negatively relative to the potential of zero charge, results in a decrease in interfacial tension and a relative flattening of the drop due to gravity. The lowered interfacial tension is caused by the repulsion between the charges in EDL.

The related content is revised in the manuscript:

“This phenomenon is caused by charge transfer across the interface, adsorption of polymer chains or ions, or other processes. Upon applying an electric field to the ink, an electrocapillary phenomenon takes place in the EDL. The charges in the EDL are influenced by the electric field, causing ions in the diffusion layer of the continuous phase and charges in the EGaIn droplet to redistribution and consequently form a charge density gradient along the direction of the electric field. The redistribution of charges leads to a change in the interfacial tension. Since a charge density gradient is formed on the surface of the droplet, this also creates a gradient in interfacial tension on the droplet surface.”

(Page16, line 311 ~ 318)

Reviewer 2 Comments:

Comment 6. Follow-up question: Since gravitational forces pressurize LM particles in the 3D structure into coalescence, does that mean 2D traces exhibit better shape retention (longer than 3 hours)?

Reply: Thank you for your valuable comment. 2D patterns indeed exhibit better shape retention, as shown in **Supplementary Fig. 18**. After 24 hours, the shapes of 2D patterns and lines do not significantly change, even in ambient conditions or at 100°C; there is no extensive merging of liquid metal droplets. In fact, the easy activation of the ink is due to the easy merging of liquid metal droplets, which also results in shape change. Therefore, even in 2D structures, the ink must be encapsulated within certain materials and cannot be directly exposed to the environment.

The related content is revised in the manuscript:

“The printed 3D objects can remain stable for up to 3 hours in ambient conditions, while 2D patterns can be preserved for over 24 hours at 100°C (Supplementary Fig. 18). The easy merging of liquid metal droplets facilitates subsequent activation of the ink's conductivity while also leading to shape changes.” (Page 12, line 233 ~ 236)

Supplementary Fig. 18 is added into the Supporting Information:

“Supplementary Fig. 18 | Morphological changes of printed objects over time. a The printed cube 3D object under ambient conditions. b The printed line at 25°C and 100°C. c The printed 2D pattern at 25°C and 100°C.”

As shown in Supplementary Fig. 18a, the printed small cube 3D object can maintain its stable shape within the first two hours. After three hours, some large EGaIn drops begin to appear from the object, indicating the rupture of the oxide layer in the ink. The printed lines and 2D patterns exhibit better shape retention. As shown in Supplementary Fig. 18b, after 24 hours, the shapes of 2D patterns and lines do not significantly change, even in ambient conditions or at 100°C. There is no extensive merging of liquid metal droplets. In fact, the easy activation of the ink is due to the easy merging of liquid metal droplets, which also results in shape change. Therefore, even in 2D structures, the ink must be encapsulated within certain materials and cannot be directly exposed to the environment.”

(Supporting information, Page 21)

Comment 7. Regarding voltage induced activation, the authors provide a good depiction of the electrocapillary force acting on a droplet in Fig 5g. However, is the activation caused by this voltage-induced motion exclusively? Could thermal expansion, via joule heating, cause a similar effect, and did the authors perform any experiments to eliminate that as a possible cause?

Reply: Thank you for your valuable comment. Firstly, we estimated the Joule heat generated during the activation process. The initial resistance of the printed line is approximately 1 M Ω . Even under a voltage of 20 V for 60 seconds, the generated Joule heat is only 0.024 J, which is insufficient to increase the line's temperature by 1°C. As shown in Figure 5e of the manuscript, at lower voltages, even if we extend the duration of the voltage applied to the printed lines to generate Joule heat continuously, it still cannot make the circuit conductive. We also placed the printed lines at 100°C for 24 hours, as shown in **Supplementary Fig. 23**. Only a small portion of the droplets on the line merged due to thermal expansion. Thus, after 24 hours at 100°C, some lines became conductive while others did not. However, as seen in **Supplement Movie 4 and Fig. 5c**, the droplet merging on the voltage-induced activated line is rapid and complete. From this, we can rule out the possibility that Joule heat caused this effect.

The related content is revised in the manuscript:

“Unlike the partial merging of droplets in ink caused by thermal expansion, this method enables rapid and complete conductivity activation of LM-HIPEG ink under ambient temperature without mechanical stimulation (Supplementary Fig. 23).” (Page15, line 302 ~ Page 16, line 305)

“Supplementary Fig. 23 / The merging of droplets on printed lines. a At 25°C after 24 hours. b At 100°C after 24 hours. c After voltage-induced activation.

*We placed the printed lines at 100°C for 24 hours, as shown in **Supplementary Fig. 23**. Only a small portion of the droplets on the line merged due to thermal expansion. Thus, after 24 hours at 100°C, some lines became conductive while others did not. The droplet merging on the voltage-induced activated line is rapid and complete, and all lines are activated under voltage induction.”* (Supporting information, Page 26)

Comment 8. In the text, it is clear that the LM-HIPEG solidifies by freezing at -30 °C. This is not as clear in Fig 3h and would be helpful to have the caption mention this.

Reply: Thank you for your valuable comment. According to your suggestion, we added annotations in Figure 3h and revised the caption in the revised manuscript.

The related content is revised in the manuscript:

“Photo of a zigzag structure solidified at -30°C supporting a weight.”

Comment 9. In Figure 2e, there are two y-axis, but it is not clear which one applies to which curve. Color coordination would be helpful here.

Reply: Thank you for your valuable comment. According to your suggestion, we have changed the colors of the curves and the axes to show their correspondence.

The related content is revised in the manuscript:

Comment 10. The following papers may be of interest to the authors:

- “Ultrastretchable, Wearable Triboelectric Nanogenerator Based on Sedimented Liquid Metal Elastomer Composite” – Pan, et al. - <https://doi.org/10.1002/admt.202000754>. This paper uses density driven sedimentation to produce a locally concentrated liquid metal elastomer composite, which becomes conductive once the liquid metal particles become densely packed.
- “Directed Assembly of Liquid Metal-Elastomer Conductors for Stretchable and Self-Healing Electronics” – Krisnadi, et al. - <https://doi.org/10.1002/adma.202001642>. Although not a printable ink, this paper uses electric field to activate a liquid metal elastomer composite.
- “Deformable liquid metal polymer composites with tunable electronic and mechanical properties” – Koh, et al. - <https://doi.org/10.1557/jmr.2018.209>. This early work on LM-elastomer composites shows a conductive inverted phase emulsion, where galinstan is the continuous phase.

Reply: Thank you for your valuable comments on our manuscript and the papers you provided. We highly appreciate them and have read the recommended papers, finding them to be of significant reference value for further understanding the rheological regulation and conductive activation of composites containing liquid metal. We believe that some of them are suitable for our research and can

be included as references.

The related content is revised in the manuscript:

“The combination of liquid state and excellent electrical conductivity make them ideal materials for electrodes in flexible electronic and printed electronic devices⁴⁻⁷.” (Page 3 26)

“The oxide layer helps to disperse LM droplets within a polymer matrix⁸, such as polyvinylalcohol (PVA) solution^{9,10}, polydimethylsiloxane¹¹⁻¹⁵.” (Page 3 36)

References:

1. Eaker, C. B. & Dickey, M. D. Liquid metal actuation by electrical control of interfacial tension. *Appl. Phys. Rev.* **3**, 031103 (2016).
2. Khan, M. R., Eaker, C. B., Bowden, E. F. & Dickey, M. D. Giant and switchable surface activity of liquid metal via surface oxidation. *Proc. Natl. Acad. Sci.* **111**, 14047-14051 (2014).
3. Frumkin, A., Polianovskaya, N., Grigoryev, N. & Bagotskaya, I. Electrocapillary phenomena on gallium. *Electrochim. Acta* **10**, 793-802 (1965).
4. Baharfar, M. & Kalantar-Zadeh, K. Emerging Role of Liquid Metals in Sensing. *ACS Sens.* **7**, 386-408 (2022).
5. Li, X. et al. A Self-Supporting, Conductor-Exposing, Stretchable, Ultrathin, and Recyclable Kirigami-Structured Liquid Metal Paper for Multifunctional E-Skin. *ACS Nano* **16**, 5909-5919 (2022).
6. Zhang, Z. X. et al. Liquid metal-created macroporous composite hydrogels with self-healing ability and multiple sensations as artificial flexible sensors. *J. Mater. Chem. A* **9**, 875-883 (2021).
7. Pan, C. F., Liu, D. Y., Ford, M. J. & Majidi, C. Ultrastretchable, Wearable Triboelectric Nanogenerator Based on Sedimented Liquid Metal Elastomer Composite. *Adv. Mater. Technol.* **5** (2020).
8. Majidi, C., Alizadeh, K., Ohm, Y., Silva, A. & Tavakoli, M. Liquid metal polymer composites: from printed stretchable circuits to soft actuators. *Flex. Print. Electron.* **7**, 013002 (2022).
9. Wang, Q., Ji, X., Liu, X., Liu, Y. & Liang, J. Viscoelastic Metal-in-Water Emulsion Gel via Host-Guest Bridging for Printed and Strain-Activated Stretchable Electrodes. *ACS Nano* **16**, 12677-12685 (2022).
10. Xu, J. et al. Printable and recyclable conductive ink based on a liquid metal with excellent surface

wettability for flexible electronics. *ACS Appl. Mater. Interfaces* **13**, 7443-7452 (2021).

11. Koh, A., Sietins, J., Slipper, G. & Mrozek, R. Deformable liquid metal polymer composites with tunable electronic and mechanical properties. *J. Mater. Res.* **33**, 2443-2453 (2018).
12. Zhou, L. y., Fu, J. z., Gao, Q., Zhao, P. & He, Y. All - printed flexible and stretchable electronics with pressing or freezing activatable liquid - metal - silicone inks. *Adv. Funct. Mater.* **30**, 1906683 (2020).
13. Neumann, T. V., Facchine, E. G., Leonardo, B., Khan, S. & Dickey, M. D. Direct write printing of a self-encapsulating liquid metal-silicone composite. *Soft Matter* **16**, 6608-6618 (2020).
14. Zhou, L.-y., Ye, J.-h., Fu, J.-z., Gao, Q. & He, Y. 4D printing of high-performance thermal-responsive liquid metal elastomers driven by embedded microliquid chambers. *ACS Appl. Mater. Interfaces* **12**, 12068-12074 (2020).
15. Jo, Y. et al. Printable Self-Activated Liquid Metal Stretchable Conductors from Polyvinylpyrrolidone-Functionalized Eutectic Gallium Indium Composites. *ACS Appl. Mater. Interfaces* **14**, 10747-10757 (2022).

Description of Additional Supplementary Files

Supplementary Movie 1: 3D printing of LM-HIPEG at a speed of 30 mm s^{-1} .

Supplementary Movie 2: Activating circuits printed on PDMS by stretching.

Supplementary Movie 3: Activating circuits by cryogenic freeze crystallization.

Supplementary Movie 4: The electrocapillary effect of EGaIn in Carbopol gel.

Activating a printed line with a 21 V voltage. Morphological changes of the line when activated under the microscope.

Supplementary Movie 5: Alternating printing of LM-HIPEG with PDMS/PTFE.

Supplementary Movie 6: Extrusion speed of LM-HIPEG under different air pressures.

Printing conditions of LM-HIPEG at different printing speeds.